# Disentangling the determinants of transposable elements dynamics in vertebrate genomes using empirical evidences and simulations

**Yann Bourgeois**[1,2]*, **Robert P. Ruggiero**[2,3], **Imtiyaz Hariyani**[2], **Stéphane Boissinot**[2]*

**1** School of Biological Sciences, University of Portsmouth, Portsmouth, United Kingdom, **2** New York University Abu Dhabi, Saadiyat Island Campus, Abu Dhabi, United Arab Emirates, **3** Department of Biology, Southeast Missouri State University, Cape Girardeau, MO, United States of America

* yann.bourgeois@port.ac.uk (YB); stephane.boissinot@nyu.edu (SB)

## Abstract

The interactions between transposable elements (TEs) and their hosts constitute one of the most profound co-evolutionary processes found in nature. The population dynamics of TEs depends on factors specific to each TE families, such as the rate of transposition and insertional preference, the demographic history of the host and the genomic landscape. How these factors interact has yet to be investigated holistically. Here we are addressing this question in the green anole (*Anolis carolinensis*) whose genome contains an extraordinary diversity of TEs (including non-LTR retrotransposons, SINEs, LTR-retrotransposons and DNA transposons). We observed a positive correlation between recombination rate and frequency of TEs and densities for LINEs, SINEs and DNA transposons. For these elements, there was a clear impact of demography on TE frequency and abundance, with a loss of polymorphic elements and skewed frequency spectra in recently expanded populations. On the other hand, some LTR-retrotransposons displayed patterns consistent with a very recent phase of intense amplification. To determine how demography, genomic features and intrinsic properties of TEs interact we ran simulations using SLiM3. We determined that i) short TE insertions are not strongly counter-selected, but long ones are, ii) neutral demographic processes, linked selection and preferential insertion may explain positive correlations between average TE frequency and recombination, iii) TE insertions are unlikely to have been massively recruited in recent adaptation. We demonstrate that deterministic and stochastic processes have different effects on categories of TEs and that a combination of empirical analyses and simulations can disentangle these mechanisms.

## Author summary

Transposable elements (TEs) are mobile DNA sequences that can replicate and insert in genomes. By doing so, they can disrupt gene function and meiotic process, but also generate evolutionary novelties. It is however unclear how different processes such as varying

---

**Data Availability Statement:** The scripts used to perform simulations using SLiM3 are available on Github (https://github.com/YannBourgeois/SLIM_simulations_TEs). All sequencing data are available

on the European Nucleotide Archive (https://www.ncbi.nlm.nih.gov/sra) under the BioProject designation PRJNA376071 (https://www.ncbi.nlm.nih.gov/bioproject/?term=PRJNA376071). All TEs counts and genotype files are available on Dryad (doi:10.5061/dryad.wpzgmsbjw).

**Funding:** This work was supported by New York University Abu Dhabi (NYUAD) research funds AD180 (to SB). The NYUAD Sequencing Core is supported by NYUAD Research Institute grant G1205-1205A to the NYUAD Center for Genomics and Systems Biology. The funding bodies had no role in designing the study, nor in data collection and interpretation.

**Competing interests:** The authors have declared that no competing interests exist.

rates of transposition, selection on TEs, linked selection and genome properties interact with each other. Here, we use the green anole (*Anolis carolinensis*) as a model, since it harbors one of the highest diversities of TEs found in a vertebrate (including non-LTR retrotransposons, LTR-Retrotransposons, DNA transposons and SINEs). By studying the population genomics of these different categories of TEs *within the same species*, we are able to disentangle processes that are specific to TE clades from general processes related to drift and selection. To do so, we use simulations of TEs in their genomic context to provide an interpretation of associations between recombination rate and statistics summarizing TE diversity and abundance. Our results highlight clear differences in TE dynamics across clades, with a clear dichotomy between SINEs/DNA-transposons and LTR-Retrotransposons/long LINEs. These differences can be mostly explained by changes in the relative impact of selection against TEs, linked selection, and insertional preferences.

## Introduction

Transposable elements (TEs) are among the genomic features that display the most variation across the living world. The nature of the interactions between these genomic 'parasites' and their hosts has likely played a considerable role in determining the size, structure and function of eukaryotic genomes [1–3]. From the perspective of TEs, genomes can be seen as an ecosystem with distinct niches. Borrowing from community ecology concepts [4,5], variation in TE composition and diversity along the genome may be due to competition for resources between clades or constraints linked to changes in environmental conditions (niche-partitioning). An alternative model would posit that TE diversity be driven by stochastic events of population size changes in the host and drift that are independent of intrinsic TE properties such as selection or transposition (neutral theory) [6]. Within a given host species, these processes can be studied through the prism of population genetics, a field that conceptually inspired the study of ecological communities. Processes linked to niche-partitioning such as varying selection against new insertions [7], variability in the use of cellular machinery and access to chromatin by different TE clades [8,9], or domestication of elements [10], may shape TEs diversity in predictable ways. On the other hand, stochastic processes at the level of individual elements, but also demography at the scale of the host [11–13], may be sufficient to explain variation in the TE landscape [4]. In addition, stochastic processes may not be constant along the genome. For example, recent investigations have highlighted the importance of recombination rates in shaping genomic diversity, due to the effects of selection at linked sites. Because of Hill-Robertson interference, regions near a selected site see their genetic diversity drop, an effect that increases in regions of low recombination [14]. This drop may not only affect nucleotide diversity, but also TEs and other structural variants.

In this work, we investigate three main factors that may impact TE distribution and diversity in the genome: direct selection on TEs, Hill-Robertson interference, and differences in their properties (e.g. preferential insertion). Many of these mechanisms make predictions about the correlation between recombination rate and diversity. For example, it is often assumed that higher recombination rates may result in higher rates of ectopic recombination, making repetitive elements more deleterious in regions of high recombination (e.g. [7,15]). This should result in negative correlations between TEs abundance/frequency and recombination. Hill-Robertson interference leads to shorter coalescence times in regions of low recombination. This may result in a faster fixation of neutral and slightly deleterious mutations, but also in lower polymorphism than in regions of high recombination [16,17]. At last, because

recombination rate is often correlated with other genomic features such as exon density, DNA repair machinery, or open chromatin, variation in TE insertion mechanisms may be reflected in correlations between their density and recombination.

In vertebrates, most of the knowledge on the micro-evolutionary dynamics of TEs is provided by studies on humans [7]. It seems clear that mechanisms such as drift, selection and migration may play an important role in shaping TEs abundance and frequencies (e.g. [11]). In addition, TEs can insert within regulatory sequences and coding regions, and have a strong potential to reduce fitness. It is therefore likely that they are under purifying selection, which should leave specific signatures such as allele frequency spectra skewed towards rare variants in TEs compared to near-neutral markers such as SNPs [18]. In human, purifying selection acting against long TEs has been demonstrated and this pattern was explained by the greater ability of long elements to mediate deleterious ectopic recombination [19]. While the human model has provided deep insights about the dynamic of LINEs in mammals, it provides only a partial picture of the dynamics of TEs as a whole, given the absence of recent activity of other categories of TEs, such as DNA transposons, in the human genome. In fact, mammalian genomes are unique among vertebrates. They are typically dominated by a single category of autonomous element, *L1*, and related non-autonomous elements (e.g. *Alu* in primates).

Non-mammalian vertebrates display a much larger TE diversity, and often include both class I elements (i.e. elements that use an RNA intermediate in their life cycle) and class II elements (i.e. elements that don't use an RNA intermediate). Class I includes LTR-retrotransposons, non-LTR retrotransposons (*i.e.* LINEs and *Penelope*) and their non-autonomous counterparts (SINEs). Class II includes a wide diversity of elements including the widespread DNA transposons. Since TEs vary in their mode of transposition, length, regulatory content and structure, it is likely that the effect they have on host fitness and how they are in turn affected by host-specific response will differ. A potentially fruitful approach to this question would be to apply the conceptual and practical tools of population genetics in a model harboring a wide diversity of active TEs. This would facilitate direct comparisons between TE categories while removing the confounding effects of host demography since all elements within the same genome share the same demographic history. The growing availability of whole-genome resequencing data, as well as the development of new computational tools, has revived the interest of the evolutionary genomics community for the analysis of TE polymorphisms within species [20,21].

Whether TEs constitute a substrate for adaptation is another area of interest. Since TEs can lead to substantial regulatory and structural variation, they may constitute targets for fast adaptation and be domesticated by the host's genome [22]. Several possible cases have now been identified at short evolutionary scales, such as the involvement of a TE insertion in industrial melanism trait in peppered moth [23], or the association between some TEs and adaptation to temperate environment or pesticides [10,24] in *Drosophila*. Identifying candidate TEs (and more generally genomic regions) for positive selection is still challenging, and requires stringent filters to keep the number of false positives at a minimum. Combining genome scans obtained from SNP data with a screening of TEs displaying strong difference in frequencies across populations should, fulfill this goal [20,25].

In this study we investigate TE variation in the green anole (*Anolis carolinensis*), which is a particularly relevant model since it is extremely diverse in terms of TE content. Its genome contains four main TE categories, each represented by multiple clades of elements: non-LTR autonomous retrotransposons (nLTR-RT; including the *L1*, *CR1*, *L2* and *Penelope* clades), SINEs, LTR-retrotransposons (LTR-RT; including the *BEL*, *Copia*, *Gypsy* and *Dirs* clade), and DNA transposons (including *hAT*, *hobo*, *Tc1/Mariner* and *helitrons* clades). There is preliminary evidence that TEs may have been involved in adaptation in anoles, for example by

inserting in the *Hox* genes cluster [26]. Previous studies have investigated patterns of genetic structure and past history: the ancestor of the green anole originally colonized Florida from Cuba between 6 and 12 million years ago [27]. A first step of divergence occurred in Florida between 3 and 2 mya (S1 Fig) [28], producing three distinct genetic clusters in Florida, the North-Eastern Florida population (NEF), the North-Western Florida population (NWF) and the South Florida population (SF), the latter being the basal one. The ancestral population of lizards now living in temperate territories diverged from the NEF cluster approximately 1 Mya. This divergence was followed by expansion northwards from Florida to the remaining South-Eastern USA, across the Gulf Coastal Plain over the last 100,000–300,000 years [29,30]. This led to the emergence of the two current northern populations, Gulf-Atlantic (GA) and Carolinas (CA). A key aspect of these studies is that they revealed large effective population sizes in all clusters, which should increase the efficiency of selection on TEs and render it easier to detect. In addition, the broad set of environmental conditions encountered by the green anole should provide opportunities for recruitment of TEs by positive selection. At last, genetic diversity is highly variable along the green anole genome, reflecting the joined effects of heterogeneous recombination rates and linked selection [30].

We take advantage of previous studies that investigated the recombination and diversity landscape along the genome to assess i) how does diversity and genomic repartition vary across different TE clades; ii) if direct selection against TE insertions is detectable; iii) how the interaction between demography, counter-selection and linked selection may impact TE frequencies and local abundance; iv) whether there is any clear evidence for positive selection acting on TEs.

## Results

### Description of polymorphic insertions

A total of 339,149 polymorphic TE insertions with no missing genotype were recovered from resequencing data obtained from 28 anoles, including the five genetic clusters identified in previous studies [29,30]. This included both reference and non-reference insertion. Two of these genetic clusters (GA and CA, referenced as Northern populations) went through a bottleneck 100,000 years ago. Note that the individual used to build the reference genome was sampled in South Carolina, which places it in the Northern populations [31]. The most abundant category of polymorphic TE found in our dataset consisted in DNA transposons (N = 132,370), followed by nLTR-RTs (N = 97,586), LTR-RTs (N = 78,472), and SINEs (N = 30,721). At a finer taxonomic scale, we mostly identified elements belonging to the *CR1*, *L2*, *L1* and *Penelope* clades for nLTR-RTs, *Gypsy* and *DIRS* for LTR-RT, and *Hobo*, *Tc1/Mariner*, *hAT* and *Helitron* for DNA transposons (Table 1). Elements such as *R4*, *RTEX*, *RTE-BovB*, *Vingi* or *Neptune* were rare and mostly fixed (Table 1), probably due to their older age. The same was observed for ancient repeats, classified as *Eulor*, *MER*, *UCON* or *REP* for DNA transposons.

### Diversity within individuals and genetic clusters

We first examined the possible impact of demography on TEs diversity and abundance. In each individual, we assessed whether heterozygous insertions were found in other green anoles or outgroups. We focused on shared heterozygosity at the individual level to better visualize intra- and inter-individual diversity (Fig 1). Singletons are more likely to be of recent origin, while heterozygous TEs shared between multiple individuals should be older, which may give information about the past and current dynamic of polymorphic elements. Given the low homoplasy of TE insertions [32,33], elements shared with the two outgroups were almost certainly found in the common ancestor, and may highlight how past demography impacted

**Table 1. Summary of TE polymorphisms in the five genetic clusters identified in the green anole, and its two Cuban counterparts.** For each cluster/outgroup, the number of polymorphic or fixed elements is given. Note that GA and CA (Northern populations) went through a bottleneck approximately 100,000 years ago.

| Category | Clade | A. allisoni | | | A. porcatus | | SF | | NWF | | NEF | | GA (Northern pop) | | CA (Northern pop) | |
|---|---|---|---|---|---|---|---|---|---|---|---|---|---|---|---|---|
| | | N | Fixed | Heterozygous | Fixed | Heterozygous | Fixed | Polymorphic | Fixed | Polymorphic | Fixed | Polymorphic | Fixed | Polymorphic | Fixed | Polymorphic |
| nLTR-RTs | CR1 | 32804 | 3892 | 2613 | 3343 | 2795 | 3557 | 6712 | 3488 | 6783 | 3328 | 13601 | 4059 | 5175 | 4357 | 3476 |
| | L2 | 26392 | 4261 | 4396 | 4469 | 5051 | 6577 | 5196 | 6604 | 4962 | 6259 | 6250 | 7004 | 2929 | 7266 | 2064 |
| | Penelope | 16208 | 978 | 1375 | 1313 | 1470 | 1074 | 2935 | 1056 | 3611 | 915 | 6426 | 1081 | 2670 | 1180 | 1876 |
| | L1 | 14181 | 1057 | 1474 | 1088 | 1594 | 1023 | 2842 | 1012 | 3231 | 914 | 5414 | 1128 | 2137 | 1243 | 1491 |
| | RTE1 | 3709 | 418 | 232 | 370 | 372 | 348 | 309 | 352 | 943 | 332 | 1152 | 362 | 606 | 375 | 221 |
| | R4 | 1516 | 166 | 345 | 253 | 427 | 265 | 308 | 274 | 310 | 286 | 243 | 303 | 134 | 301 | 98 |
| | RTE_BovB | 920 | 240 | 209 | 302 | 235 | 358 | 167 | 377 | 151 | 347 | 187 | 380 | 83 | 392 | 66 |
| | Vingi | 860 | 360 | 174 | 306 | 151 | 777 | 75 | 783 | 77 | 758 | 102 | 823 | 37 | 826 | 34 |
| | RTEX | 496 | 128 | 134 | 206 | 146 | 416 | 73 | 425 | 68 | 380 | 116 | 447 | 49 | 465 | 31 |
| | Neptune | 376 | 14 | 4 | 4 | 12 | 6 | 37 | 4 | 57 | 3 | 215 | 3 | 56 | 4 | 28 |
| | Other | 124 | 14 | 18 | 18 | 24 | 25 | 22 | 26 | 18 | 24 | 45 | 25 | 28 | 28 | 15 |
| DNA transposons | Hobo | 45421 | 1380 | 6693 | 986 | 6900 | 244 | 11869 | 31 | 13042 | 2 | 19344 | 6 | 9900 | 39 | 6509 |
| | Tc1/Mariner | 37718 | 8380 | 6692 | 8072 | 8133 | 4533 | 6681 | 4600 | 6386 | 4190 | 11070 | 4759 | 4661 | 4935 | 3112 |
| | hAT | 25165 | 3520 | 5115 | 6705 | 5043 | 8679 | 5348 | 8778 | 5115 | 7841 | 7515 | 9058 | 4252 | 9461 | 2797 |
| | Helitron | 19266 | 1730 | 2008 | 1093 | 2491 | 147 | 3899 | 21 | 5007 | 2 | 7779 | 4 | 2424 | 24 | 1499 |
| | Other | 3517 | 569 | 1015 | 1783 | 878 | 2847 | 636 | 2958 | 554 | 2666 | 850 | 3098 | 419 | 3203 | 314 |
| | Chapaev | 1229 | 154 | 329 | 491 | 284 | 783 | 286 | 833 | 204 | 728 | 360 | 837 | 198 | 896 | 123 |
| | MER | 17 | 6 | 3 | 8 | 4 | 13 | 4 | 14 | 3 | 12 | 5 | 16 | 1 | 15 | 2 |
| | Eulor | 16 | 2 | 5 | 10 | 3 | 13 | 3 | 13 | 3 | 13 | 3 | 15 | 1 | 15 | 1 |
| | UCON | 11 | 0 | 1 | 4 | 5 | 7 | 4 | 9 | 2 | 9 | 2 | 10 | 1 | 10 | 1 |
| | Chompy | 5 | 1 | 2 | 3 | 1 | 4 | 1 | 5 | 0 | 5 | 0 | 4 | 1 | 5 | 0 |
| | Harbinger | 3 | 2 | 1 | 0 | 1 | 1 | 2 | 2 | 1 | 3 | 0 | 2 | 1 | 3 | 0 |
| | REP | 2 | 0 | 1 | 1 | 1 | 1 | 1 | 2 | 0 | 1 | 1 | 1 | 0 | 2 | 0 |
| LTR-RTs | Gypsy | 45625 | 1037 | 1223 | 985 | 1299 | 1338 | 7408 | 1029 | 9186 | 940 | 15157 | 1106 | 12939 | 1218 | 8020 |
| | Other LTRs | 13946 | 219 | 349 | 241 | 322 | 753 | 3719 | 441 | 4237 | 366 | 6215 | 419 | 2958 | 509 | 2153 |
| | BEL | 9391 | 183 | 189 | 125 | 234 | 166 | 1329 | 157 | 1154 | 148 | 4380 | 175 | 1929 | 194 | 1369 |
| | Dirs | 6873 | 391 | 577 | 297 | 607 | 362 | 1226 | 320 | 1618 | 315 | 1796 | 363 | 1332 | 393 | 930 |
| | Copia | 1962 | 324 | 268 | 315 | 550 | 257 | 279 | 239 | 392 | 233 | 465 | 256 | 233 | 276 | 132 |
| | ERV | 674 | 79 | 62 | 80 | 63 | 92 | 112 | 89 | 135 | 87 | 178 | 94 | 130 | 102 | 75 |
| | Ultra-conserved | 1 | 0 | 0 | 0 | 1 | 1 | 0 | 1 | 0 | 0 | 1 | 1 | 0 | 1 | 0 |
| SINEs | SINE2 | 20716 | 4121 | 5185 | 4568 | 5757 | 5600 | 3275 | 5661 | 2954 | 5331 | 3846 | 5837 | 1769 | 5960 | 1294 |
| | Non-assigned (ACASINE) | 4802 | 997 | 1577 | 1427 | 1564 | 1908 | 818 | 1941 | 836 | 1839 | 1024 | 1984 | 554 | 2024 | 418 |
| | SINE1 | 4115 | 271 | 440 | 195 | 569 | 57 | 1031 | 27 | 1030 | 23 | 1423 | 25 | 473 | 29 | 367 |
| | SINE3 | 1083 | 87 | 365 | 297 | 380 | 500 | 191 | 518 | 173 | 463 | 243 | 536 | 115 | 549 | 84 |
| | MIR-like | 5 | 0 | 2 | 4 | 1 | 5 | 0 | 5 | 0 | 2 | 3 | 5 | 0 | 5 | 0 |

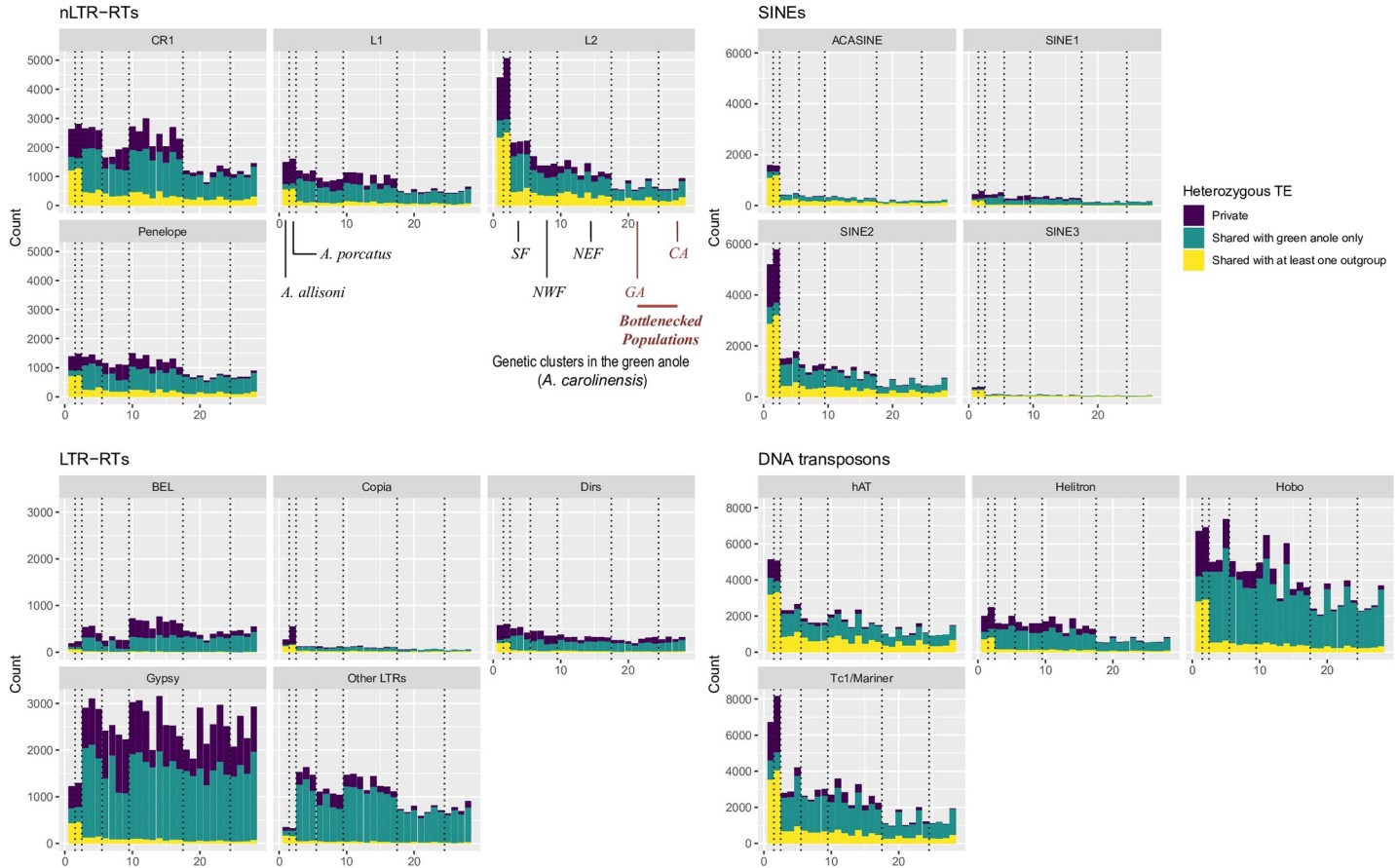

**Fig 1. Count of heterozygous sites across all 28 individuals included in this study.** Vertical dotted lined delimit the five main genetic clusters and the two outgroups in this order: *A. allisoni* and *A. porcatus*, SF, NWF, NEF, GA and CA. See S1 Fig for more details about these clusters.

individual TE landscapes. An examination of the repartition of polymorphic insertions across individuals showed a similar pattern across nLTR-RTs, SINEs and DNA transposons. On average, more heterozygous TEs were observed in individuals from the Floridian populations, which became established about two million years and remained stable and large (effective population size, $N_e$ ~ 1 million) since colonization from Cuba. For these three categories, heterozygous TEs (private or shared) are more abundant in the outgroups (which correspond to the 2 Cuban anole species) and in the Floridian populations but become rarer in populations that expanded out of Florida, which is consistent with the loss of genetic variation experienced in those more recently established populations. In addition, for the most abundant clades, there were always more fixed insertions in GA and CA than in Floridian populations with similar sample sizes (Table 1). These patterns are consistent with drift leading to faster fixation or elimination of polymorphic TEs. For nLTR-RTs and SINEs, *L2* and *SINE2* elements displayed a large number of heterozygous TEs found only in the two outgroups, but also displayed a large proportion of heterozygous sites shared between *A. carolinensis* and either *A. porcatus* or *A. allisoni*. The same was observed for the DNA transposons *Tc1/Mariner* and *hAT*. This suggests that a substantial proportion of elements inserted before the split between these species, and that drift may have led to gradual loss of shared elements. *Hobo*, *Helitron*, *SINE1*, *L1*, *CR1* and *Penelope* maintained a relatively high proportion of private insertions in individuals from Florida, less shared heterozygous sites and similar number of heterozygous insertions when

compared to the outgroups. This is consistent with elements at lower frequencies in the common ancestor, either because of stronger purifying selection or more recent transposition activity, leading to less shared variation between present genetic groups and species.

On the other hand, for LTR-RTs, elements from the *Gypsy* and *BEL* clades displayed a large number of private insertions in the green anole, with many insertions found only in a single individual, and no clear pattern of reduced abundance in bottlenecked populations from the Northern cluster. This can be interpreted as a signature of recent and active transposition in the green anole lineage. This was especially clear for *Gypsy* elements, suggesting a burst of transposition following colonization from Cuba.

A visual inspection of allele frequency spectra (AFS) confirmed the effect of demography on TEs (Fig 2, S2–S5 Figs): for DNA transposons, nLTR-RTs and SINEs, spectra were skewed toward singletons in genetic clusters with large population sizes (SF, NEF, NWF), while this trend was less pronounced in clusters having been through a recent bottleneck (GA and CA). This was reflected by systematically higher average allele frequencies in GA than in NEF (Wilcoxon tests, $P < 5.7 . 10^{-12}$ except for *SINE3*; $P = 0.03$), the only exception being ACASINE for

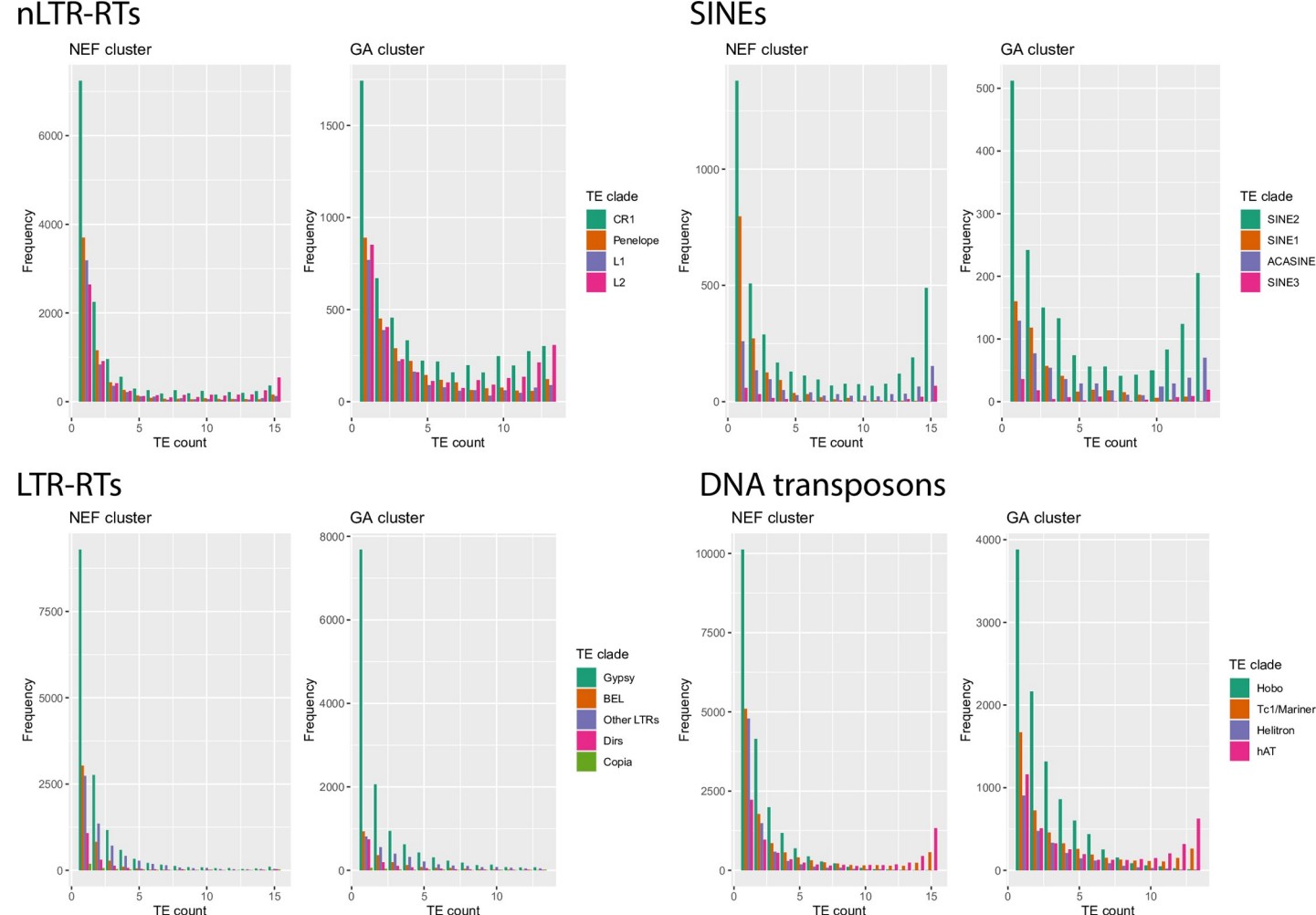

**Fig 2. Allele frequency spectra for TEs belonging to two genetic clusters identified in the green anole.** NEF (N = 8 diploid individuals) corresponds to a large, stable population from Florida, and GA (N = 7 diploid individuals) corresponds to a more recently established population having colonized northern environments in the last 100,000 years.

which no significant difference was observed. This is consistent with the excess of frequent alleles expected in the case of population contraction. These differences were however less clear for LTR-RTs, with spectra strongly skewed towards singletons in all populations. While AFS were clearly U-shaped for the other three types of elements, almost no LTR-RT insertion was found at very high frequencies. Such a pattern is consistent with recent activity and purifying selection preventing insertions to reach high frequencies. There were also differences within different types of elements. For non-LTR retrotransposons, elements such as *Poseidon* or *RTEBovB* were mostly found at high frequencies (Table 1). Elements such as *RTE1*, *L1*, *CR1* and *Penelope* displayed a stronger skew towards singletons than *L2*. In SINEs, *SINE1* had more singletons, while other elements were more frequent. For DNA transposons, the skew towards singletons was strongly pronounced for *Hobo* and *Helitron*, and very few fixed insertion were found (Table 1), suggesting either stronger purifying selection or a recent increase in transposition rate.

## Correlation of TE density with recombination and differentiation reveals discordant patterns

Studies focusing on SNPs have revealed that regions of low recombination display lower diversity and stronger differentiation between populations due to the effects of linked selection [34,35]. First, we tested whether typical signals of linked selection could be observed along the genome by examining correlations between recombination rates, derived allele frequencies, and absolute ($d_{XY}$) and relative ($F_{ST}$) measures of differentiation computed over SNP data in non-overlapping 1Mb windows (Fig 3). We focused on the six main autosomes of the green anole. If linked selection shapes genomic diversity along the genome, there should be 1) positive correlations between diversity indices (average derived allele frequency, $d_{XY}$) and

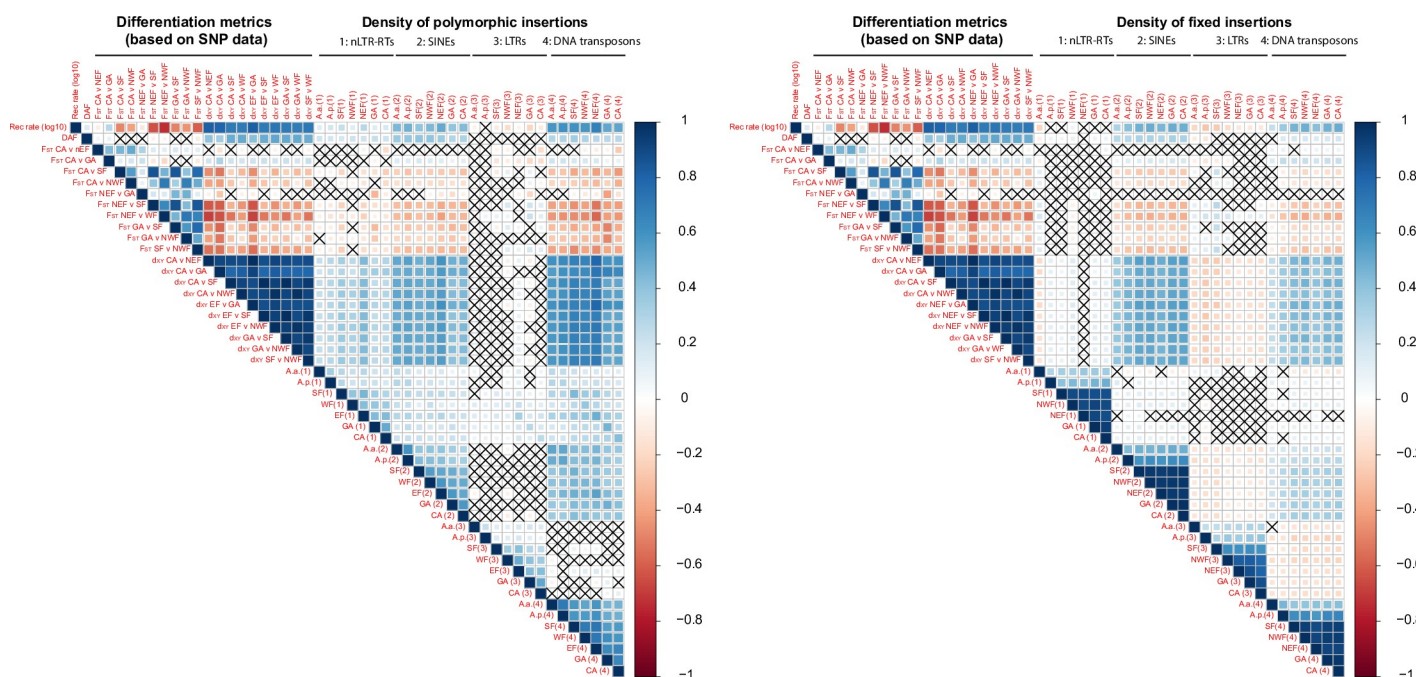

**Fig 3. Correlograms illustrating Spearman's rank correlation coefficients between TE densities in 1 Mb windows and SNP-based statistics such as recombination rate (measured as $r/\mu$, see Methods), pairwise relative ($F_{ST}$) and absolute ($d_{XY}$) measures of differentiation, and derived SNP frequency in the NEF cluster (DAF).** Correlations with $P>0.05$ are indicated with a cross.

recombination, 2) negative correlations between differentiation measures ($F_{ST}$) and recombination, 3) consistency in genomic regions displaying high or low values for $F_{ST}$ or $d_{XY}$ across all pairwise comparisons. This is in line with our observations, with mostly positive correlations between recombination rate, diversity and absolute divergence for all pairwise comparisons between the five genetic clusters (Fig 3). Pairwise relative measures of differentiation ($F_{ST}$) were negatively correlated with recombination rate, $d_{XY}$, and derived allele frequencies, which is consistent with a role of linked selection reducing diversity in regions of low recombination across all genetic clusters. Indices of differentiation comparing CA or GA with other populations were less correlated with indices of differentiation estimated between pairs of clusters from Florida, suggesting a role for recent expansion in blurring the expected correlations.

Then, we examined densities of polymorphic and fixed TEs across four main categories of TEs (Fig 3). Assuming they are nearly neutral, linked selection should have a similar effect on TEs as on SNPs. The stronger Hill-Robertson interference observed in regions of low recombination should lead to a lower number of polymorphic TEs there. On the other hand, it is generally assumed that rates of ectopic recombination increase with crossover rates. In that case, elements involved in ectopic recombination should be under strong purifying selection, slowing the accumulation of TEs in regions of high recombination compared to regions of low recombination. This should result in decreasing densities of both polymorphic and fixed elements as recombination increases. TE densities were positively correlated with recombination rate, diversity and relative measures of differentiation for SINEs and DNA transposons. Correlations were weaker for nLTR-RTs, and almost absent for LTR-RTs. The density of fixed LTR-RTs even followed an opposite pattern, with more fixed insertions in regions of low recombination and high $F_{ST}$. For fixed nLTR-RTs, correlations were weak or absent. This suggests that purifying selection against LTR-RT and to some extent nLTR-RTs may explain the variation in their local abundance and diversity.

The lower abundance of some TE categories in regions of low recombination was not explained by a higher density of functional elements that could increase their deleterious effects (S6 Fig). Exon density was positively correlated with recombination rate (Spearman's $rho = 0.15$; $P = 9.1.10^{-7}$), which suggests that regions of high recombination may also be more frequently transcribed, and are therefore more often in an open chromatin state.

TE densities were positively correlated with each other across hosts' populations for all TEs, with correlations strengthening as comparisons involved more closely related pairs of populations. This effect is expected due to a longer shared history for related genetic clusters.

## Comparison of TE diversity across TE clades in a demographically stable genetic cluster

We assessed whether purifying selection had a direct impact against TEs by examining average TE frequencies in 1Mb windows and comparing it to the frequencies of derived SNPs. To obtain a more accurate estimate of frequency, we focused on the population with the largest sample size and with a historically stable effective population size, NEF [30]. We also examined diversity at the clade level to highlight specific dynamics. We excluded TE clades with less than 5000 elements (Table 1), and merged SINEs that were not *SINE2* together to provide a comparison within the category. We examined these statistics for SNPs and the main clades within the four main TE categories (Fig 4). Average TE frequencies were lower for LTR-RTsthan for SNPs and the differences were statistically significant (frequencies of 0.10, 0.15, 0.13, 0.17, and 0.26 for *BEL*, *Dirs*, *Gypsy*, unclassified LTRs and derived SNPs respectively; paired-samples Wilcoxon tests, all $P<2.2.10^{-16}$) across all clades. This is consistent with either purifying selection against these elements, and/or their younger age. The same was observed for *CR1*, *L1* and

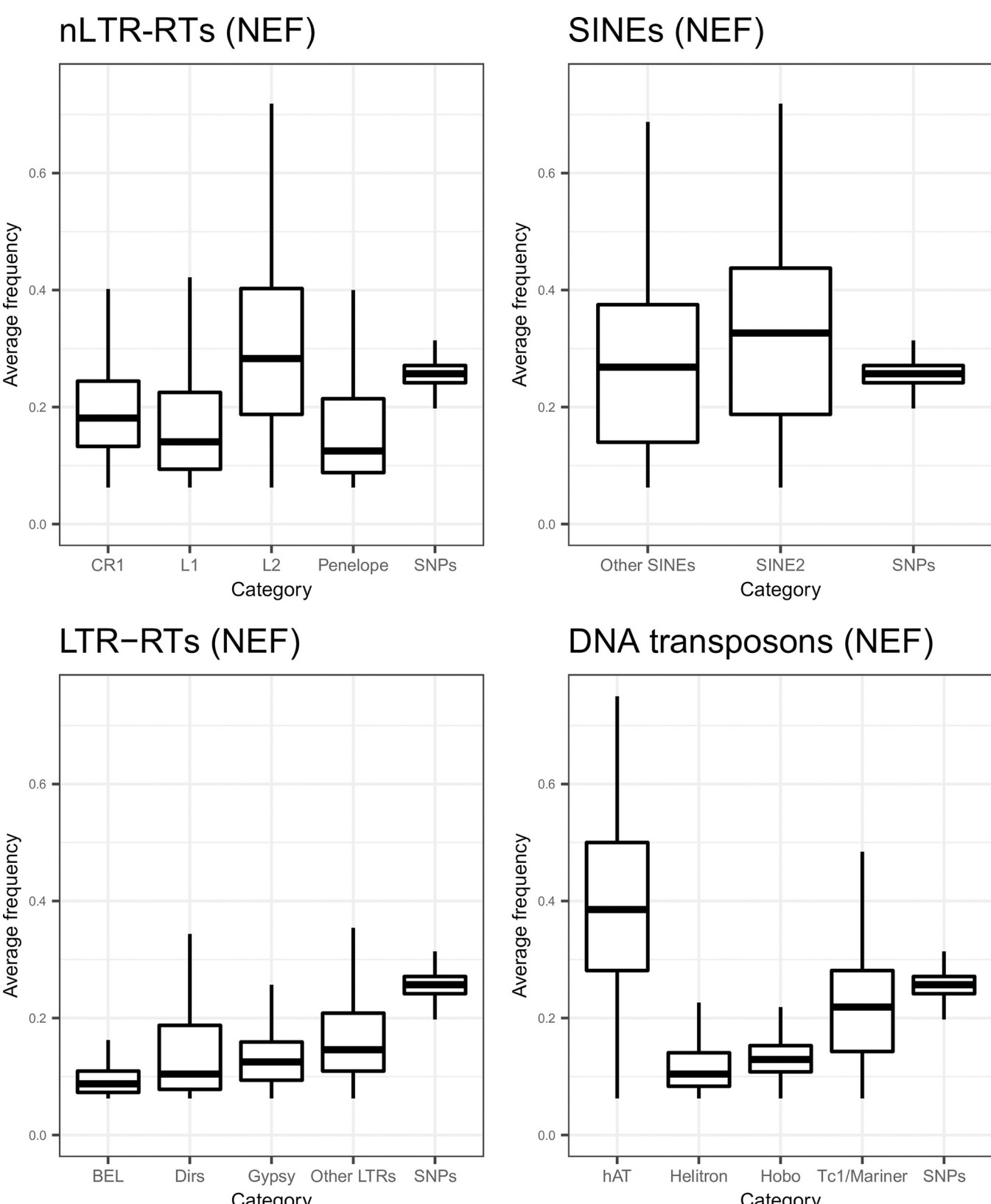

**Fig 4. Boxplots of average TE frequency for each main TE category in the NEF population.** For SNPs, the derived allele frequency was obtained by assigning variants to ancestral and derived states using *A. allisoni* and *A. porcatus*.

*Penelope* (frequencies of 0.19, 0.17 and 0.16), but not *L2* (frequency of 0.30), for which the average frequencies were significantly higher than derived SNPs (all $P<1.10^{-11}$). The average frequency of SINEs other than *SINE2* was 0.28, not substantially different from SNPs ($P = 0.88$), and was even higher for SINE2 (0.33, $P = 5.5.10^{-12}$). For DNA transposons, *Hobo*, *Helitron*, and to a lesser extent, *Tc1/Mariner* displayed lower frequencies than SNPs (0.13, 0.12 and 0.22 respectively, all $P<2.2.10^{-16}$). On the other hand, *hAT* displayed an average frequency of 0.39, substantially higher than SNPs ($P<2.2.10^{-16}$). Elements at a higher frequency than derived SNPs are likely ancient, and their high frequency is best explained by a non-equilibrium dynamic, with a lack of recent transposition resulting in a depletion in the lower frequencies of the allele frequency spectrum. Because DNA transposons replicate through a cut-and-paste mechanism, it may happen that some insertions be removed from a given insertion site. Nevertheless, the large effective population sizes considered here would make any substantial impact of occasional cut-and-paste extremely limited in terms of allele frequency.

TEs involved in ectopic recombination should be subject to purifying selection, becoming stronger in regions of high recombination. In addition, the higher exon density in these regions (S6 Fig) may increase the odds that these TEs alter gene expression. This should result in reduced frequency of polymorphic insertions and abundance of elements in regions of high recombination and high gene density. To test whether TEs from different clades followed this predicted pattern, we assessed whether their average frequency, density of polymorphic insertions, and density of fixed insertions, varied with the recombination rate (Figs 5, 6 and 7, Table 2). For all LTR-RTs, we observed negative correlations between recombination rate and average frequency (Fig 5). Weak, negative correlations were also observed when replacing frequency by the density of fixed insertions (Fig 7), the strongest trend being observed with

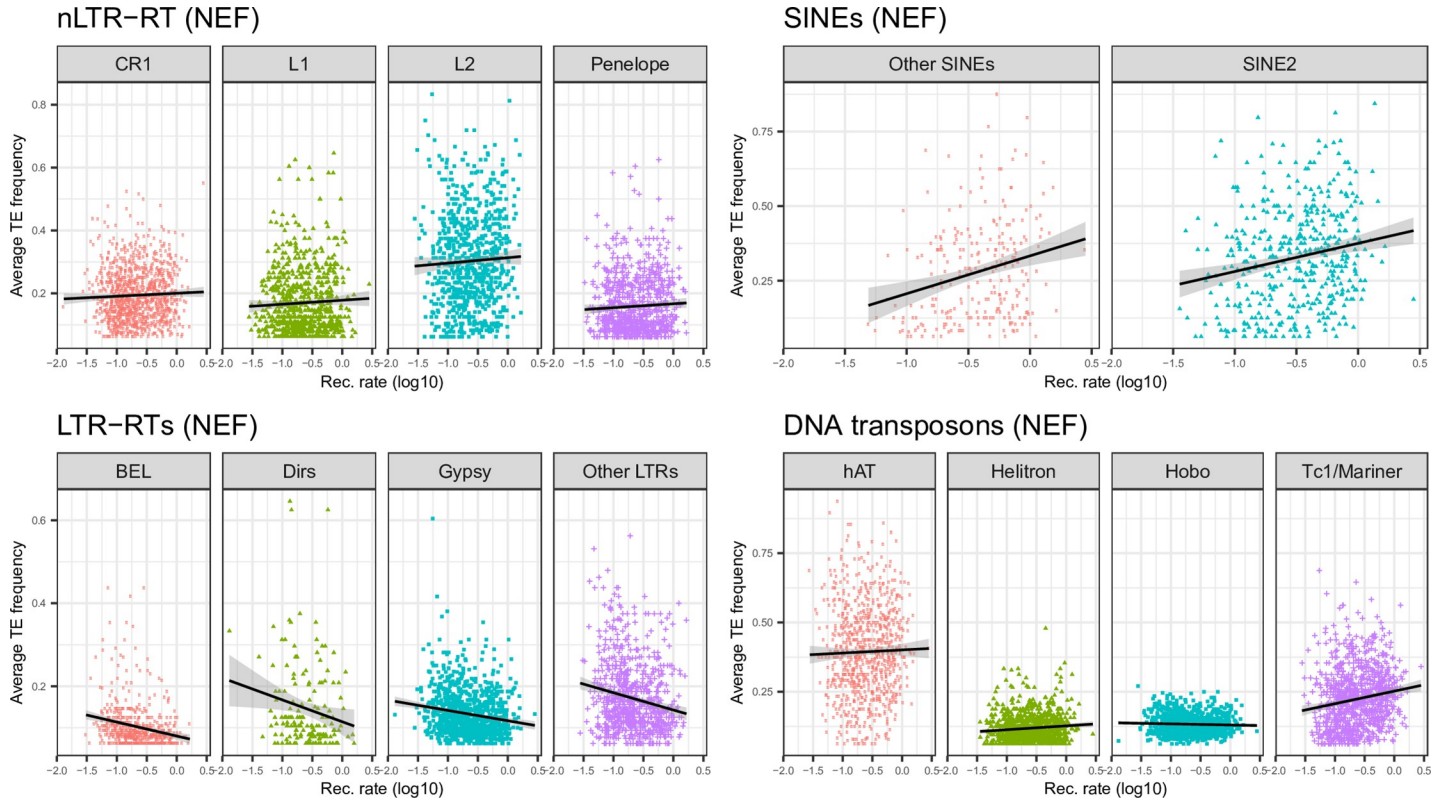

**Fig 5. Plots of average TE frequency against recombination rate computed over 1Mb windows for each main TE clade in the NEF population.**

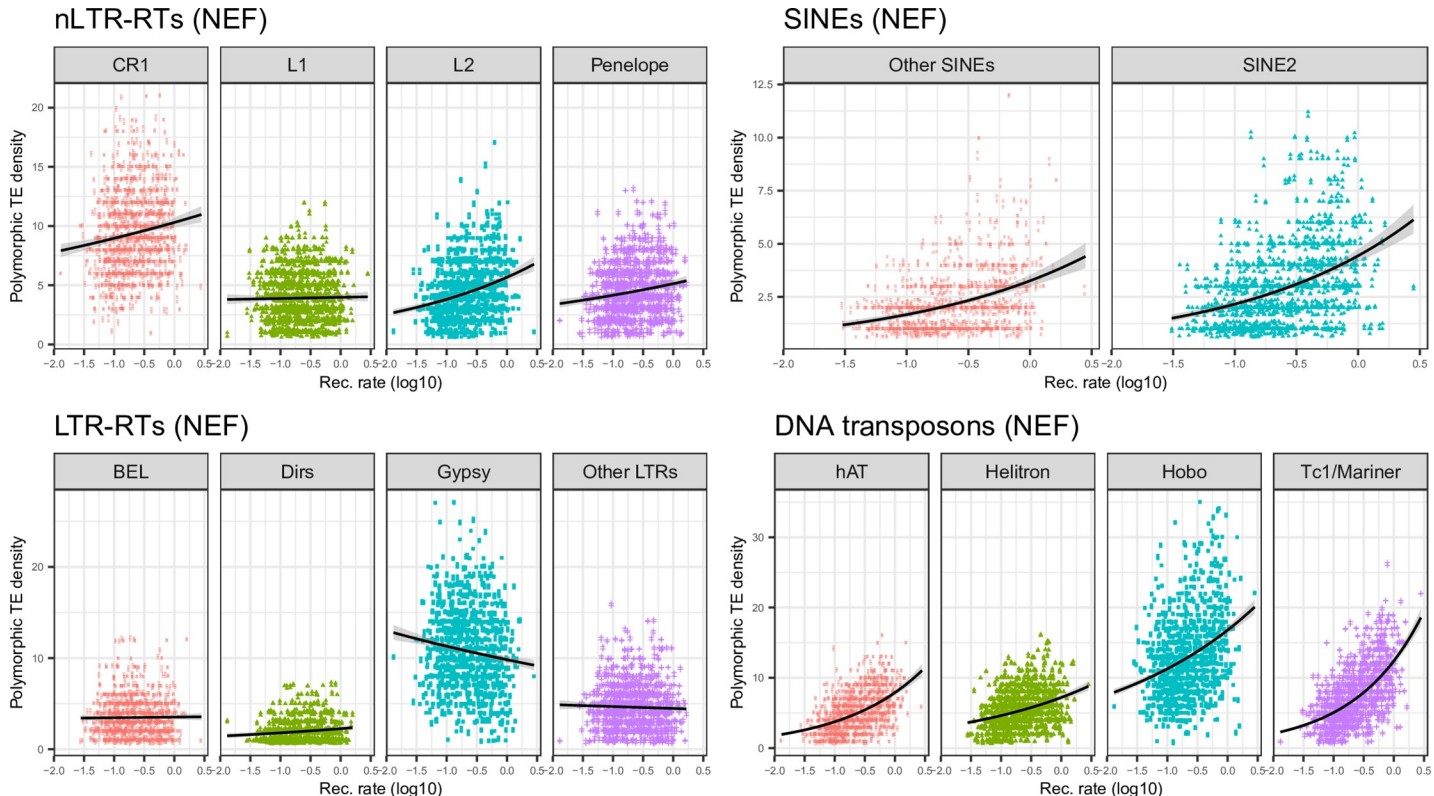

**Fig 6. Plots of polymorphic TE density against recombination rate computed over 1Mb windows for each main TE clade in the NEF population.**

*Gypsy*. For the latter, negative correlation between the density of polymorphic sites and recombination was observed (Fig 6). This pattern is clearly consistent with a stronger deleterious effect of these elements in regions of high recombination and gene density. Correlations were however weak (*BEL*, unclassified LTR-RT) for other LTR-RTs. They were significantly positive for *Dirs*. SINEs and DNA transposons (except *Hobo*) showed positive correlations between all three summary statistics and recombination rate, which may be partly explained by linked selection and a lack of strong purifying selection. For *Hobo*, the only significant correlation was found between recombination rate and the density of polymorphic sites, probably because of the rather low number of fixed insertions, obscuring correlations.

For nLTR-RTs, we did not observe significant correlations between recombination and TE frequency or the density of fixed insertions, except for *CR1* (Figs 5 and 7; Table 2). Positive correlations were however observed for *Penelope*, *CR1* and *L2* when examining the density of polymorphic sites. We however suspect that this lack of clear correlation may be due to variation in the strength of purifying selection among nLTR-RTs. Previous studies in vertebrates and *Drosophila*, [7,13,36,37] have shown that the effects of TE insertions on fitness may be correlated with their length. This may be due to the fact that the odds of homologous recombination rise with the length of homologous fragments [38], or because longer elements contain promoter sequences that may have more deleterious effects on nearby genes. Truncation in LINEs occurs at the 5' end of elements, which makes MELT estimates of their length accurate since it detects TEs based on reads mapping the ends of the insertion. To assess whether purifying selection acted more strongly on longer elements, we examined the correlation between recombination rates and the average length of fixed and polymorphic LINEs (which make

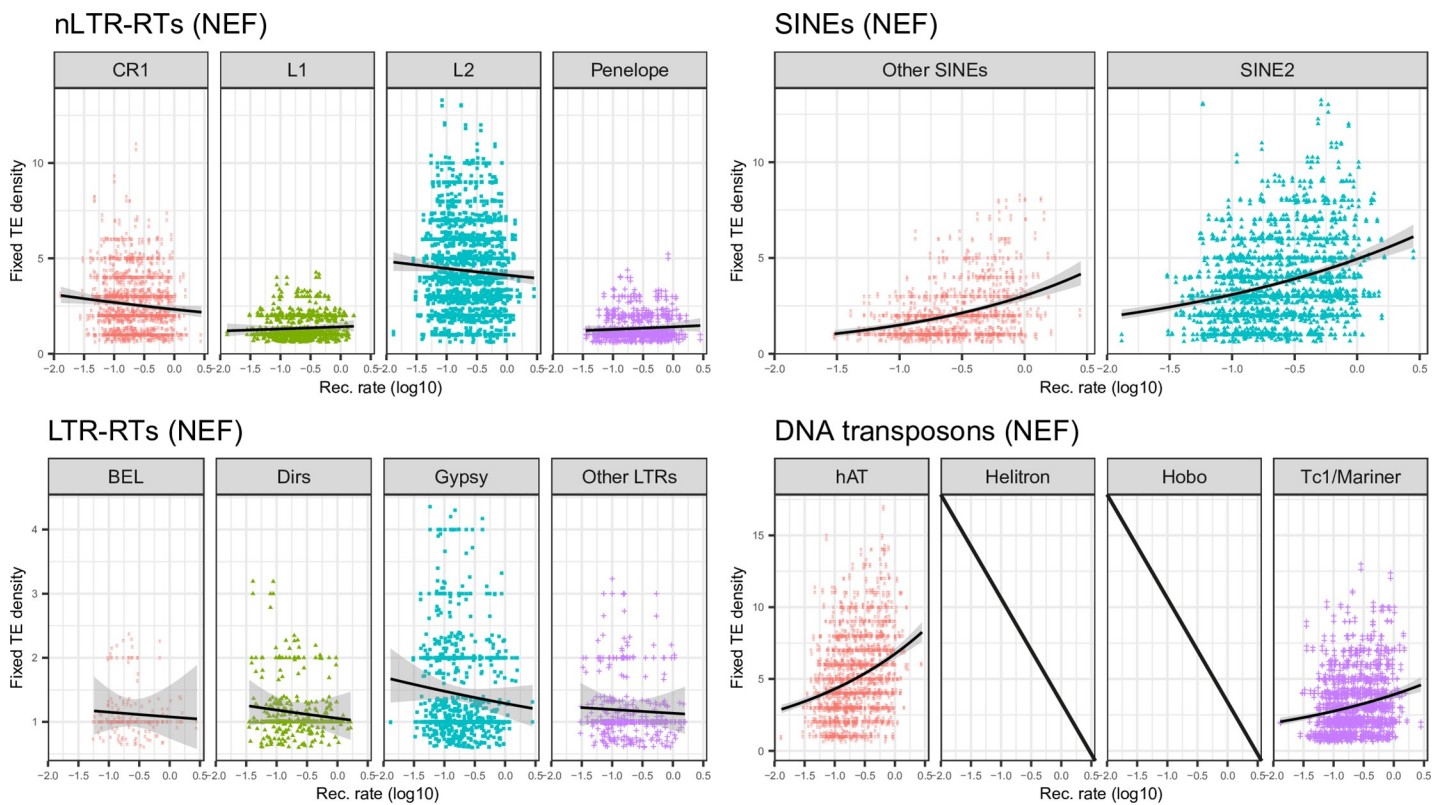

**Fig 7. Plots of fixed TE density against recombination rate computed over 1Mb windows for each main clade in the NEF population.** For *Helitron* and *Hobo*, there are not enough fixed insertions.

most of nLTR-RTs but exclude *Penelope*) in 1Mb windows (Fig 8), and observed a clear negative correlation between these two statistics (Spearman's *rho* = -0.16, -0.26, -0.21 for *CR1*, *L1* and *L2* respectively, $P<5.10–7$). LINEs that were fixed in the NEF population were also shorter than the polymorphic ones. We then focused on short LINEs (<20% of the maximum length of their respective clade) to assess whether they were also erased from regions of high recombination. We used 10Mb windows to increase the number of insertions and avoid losing too much information. We then reexamined the correlations between recombination rate and our three summary statistics (Fig 8). We found a positive correlation between frequency and recombination rate for short *CR1* and short *L2*. All short elements showed positive correlations between recombination and the density of polymorphic elements, while no clear correlation was observed for the density of fixed elements (Table 2, Fig 8). For long LINEs (>30% of the maximum length of their specific clade), we observed strong negative correlations between TE frequency, the density of fixed insertions, and recombination. The same was observed with the density of polymorphic insertions, except for *L2* (Table 2). These results suggest that weak correlations observed at the scale of the whole clade are explained by non-uniform, length-dependent selection against the elements. Short LINEs are therefore more likely under the influence of linked selection, while long LINEs display patterns that are closer to observations in LTR-RT, suggesting a stronger influence of purifying selection.

These results could also be explained by a higher rate of deletion in TE insertions located in regions of high recombination [39], or older elements containing more deletions. However, an examination of the start and end coordinates of insertions on their consensus did not reveal any substantial truncation at the 3' end (S7 Fig), which would be expected if deletion occurred

**Table 2. Summary of correlations observed between average recombination rate, the average frequency of TEs, the density of polymorphic TEs and the density of fixed elements.** For short and long nLTR-RTs, due to the low number of fixed insertions in 1Mb windows, we present results for 10MB windows instead. The last column provides an interpretation of the correlations obtained in simulations and observed in empirical data. For simulated TEs, we distinguish between outcomes where TEs are at high frequency (higher than SNPs) and low frequency (lower than SNPs). Pur. Selec. Ect. Rec.: Purifying Selection against ectopic recombination; Linked Sel.: Linked Selection; Pref. Ins.: Preferential Insertion in regions of high recombination/open chromatin; Anc. Burst: Ancient Burst of Transposition; (): the process may occur but does not impact the direction of correlations (for simulations), or is possible but no conclusive evidence is provided by the three summary statistics (for empirical observations). NA: for *Helitron* and *Hobo*, the lack of fixed insertions prevents the computation of these statistics. *: $P$-value<0.05; **: $P$-value<0.01; ***: $P$-value<0.001.

| Category | Superfamily/simulation | Correlation between recombination rate and: | | | |
|---|---|---|---|---|---|
| | | Average frequency | Polymorphic density | Fixed density | Dominant process |
| **simulations** | simulated TE | + | + | + | Linked Sel. + Pref. Ins. |
| | simulated TE | + | + | - | Linked Sel. |
| | simulated TE (high frequency) | - | + | + | Linked Sel. + Anc. Burst + Strong Pref. Ins. |
| | simulated TE (high frequency) | - | + | - | Linked Sel. + Anc. Burst |
| | simulated TE (low frequency) | - | + | - | Pur. Selec. Ect. Rec. + (Anc. Burst) + Pref. Ins. |
| | simulated TE | - | - | - | Pur. Selec. Ect. Rec. + (Anc. Burst) |
| **nLTR-RTs** | *CR1* | 0.05 | 0.15*** | -0.08 * | Mixture |
| | *CR1* (short) | 0.30** | 0.25** | -0.24* | Linked Sel. |
| | *CR1* (long) | -0.28 ** | -0.14 | -0.32 ** | Pur. Selec. Ect. Rec. |
| | *L1* | 0.02 | 0.01 | 0.08 | Mixture |
| | *L1* (short) | -0.006 | 0.35 *** | 0.08 | Linked Sel. + Pref. Ins.? |
| | *L1* (long) | -0.25 * | -0.29 ** | -0.57 *** | Pur. Selec. Ect. Rec. |
| | *L2* | 0.05 | 0.29 *** | -0.06 | Mixture |
| | *L2* (short) | 0.21 * | 0.50 *** | -0.10 | Linked Sel. + (Pref. Ins.) |
| | *L2* (long) | -0.24 * | 0.02 | -0.32 ** | Pur. Selec. Ect. Rec. + (Pref. Ins.) |
| | *Penelope* | 0.04 | 0.15*** | 0.08 | Linked Sel. + Pref. Ins.? |
| **SINEs** | *SINE2* | 0.19 *** | 0.42 *** | 0.28 *** | Linked Sel. + Pref. Ins. |
| | Other SINEs | 0.24 *** | 0.37 *** | 0.35 *** | Linked Sel. + Pref. Ins. |
| **LTR-RTs** | *Dirs* | -0.21 *** | 0.14*** | -0.09 | Pur. Selec. Ect. Rec. + Pref. Ins. |
| | *BEL* | -0.28 *** | 0.01 | -0.05 | Pur. Selec. Ect. Rec. + (Pref. Ins.) |
| | *Gypsy* | -0.18 *** | -0.13 *** | -0.11 * | Pur. Selec. Ect. Rec. |
| | Other LTRs | -0.18 *** | -0.04 | -0.04 | Pur. Selec. Ect. Rec. + (Pref. Ins.) |
| **DNA transposons** | *hAT* | 0.04 | 0.51 *** | 0.29 *** | Linked Sel. + Pref. Ins. + Anc. Burst |
| | *Helitron* | 0.10 ** | 0.32 *** | NA | Linked Sel. + (Pref. Ins.) |
| | *Hobo* | -0.03 | 0.32 *** | NA | Linked Sel. + (Pref. Ins.) |
| | *Tc1/Mariner* | 0.20 *** | 0.59 *** | 0.20 *** | Linked Sel. + Pref. Ins. |

once the element is already inserted. We only observed truncation at the 5' end, which is consistent with truncation during the insertion process.

## Simulations clarify the relative impact of purifying selection, linked selection and bursts of transposition on autonomous retrotransposons diversity

Our results reveal many combinations of correlations between TE diversity and recombination rate. To clarify and illustrate the conditions under which these combinations arise, we built a simple model of retrotransposon evolution in the forward-in-time simulator SLiM3 [40]. We simulated a 4Mb fragment with two recombination rates and negative selection on 10% of the non-coding SNPs. Recombination was high on the first and last Mb, and low for the 2Mb in the middle of the fragment. To reflect varying density of functional sites between regions of low and high recombination (S6 Fig), the density of coding sequences was 10,000 bp/Mb for the 2Mb in the middle of the fragment and 20,000 bp/Mb for the first and last Mb. Coding

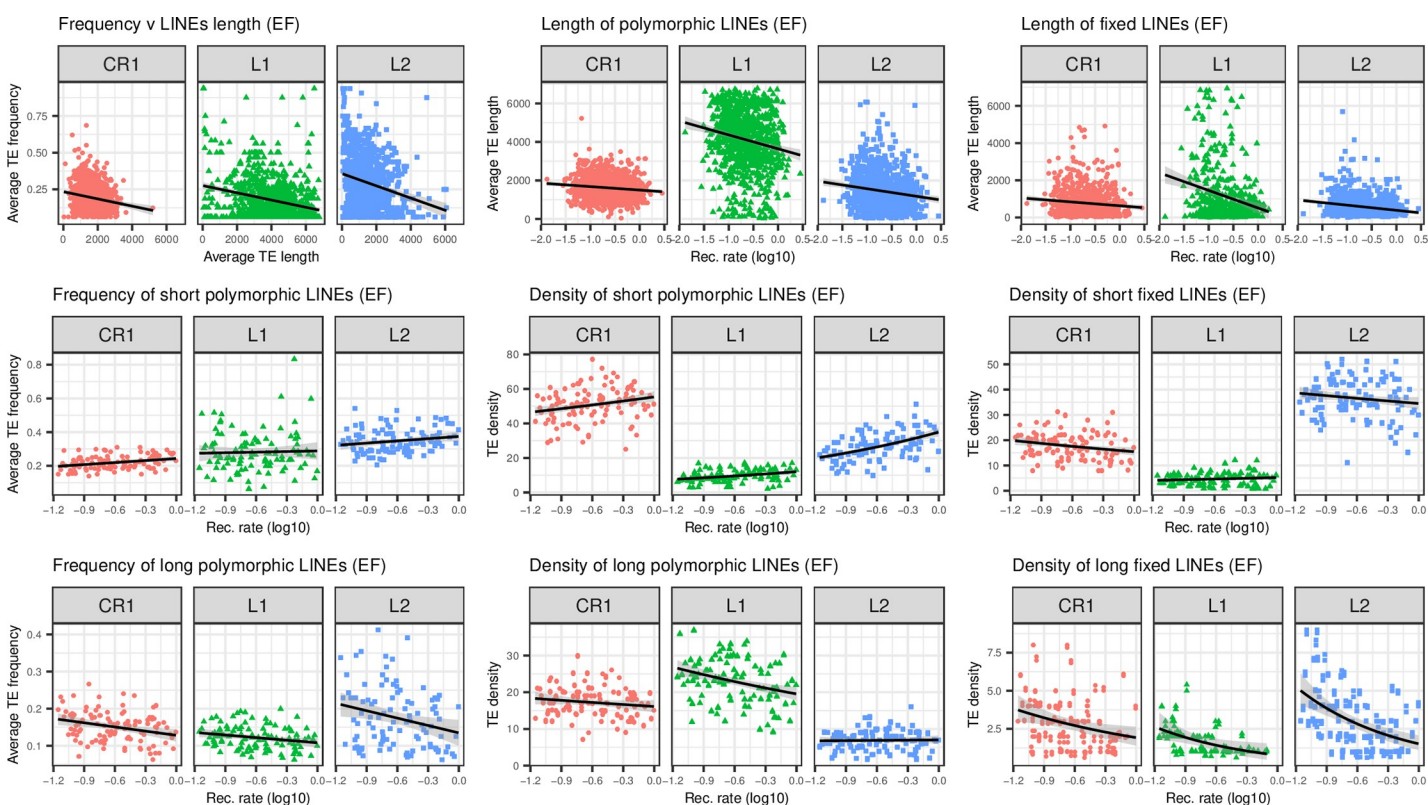

**Fig 8.** Top: plots of LINEs (i.e. nLTR-RTs excluding *Penelope*) length against recombination rate. Middle: Plots of average frequency, density of polymorphic insertions and density of fixed insertions for short LINEs, Bottom: same as middle row, for long LINEs. For middle and bottom plots, average frequencies and densities are computed for 10Mb windows.

sequences included negative selection on 70% of new mutations. Two categories of TEs were simulated, "short" TEs that were weakly deleterious (Fig 9, blue boxplots), and "long" TEs (red boxplots) that were more deleterious in regions of high recombination. Both long and short TEs falling in coding regions were strongly deleterious, with a selection coefficient $s$ = -1. We then examined the same three summary statistics than earlier: the average frequency of polymorphic insertions, the density of polymorphic insertions, and the density of fixed insertions (Fig 9). Short TEs showed higher average frequencies in regions of high recombination when transposition was kept constant, a pattern consistent with expectations if linked selection increases lineage sorting in regions of low recombination (Fig 9, panels A). This trend was however reversed if transposition occurred as a single ancient burst (panels B). In that case, average TE frequencies were also higher, due to the older age of insertions. Moreover, because linked selection leads to faster lineage sorting in regions of low recombination, polymorphic insertions that survive after the burst reach higher frequencies, explaining the observed correlation. On the other hand, long TEs displayed lower average frequencies in regions of high recombination, due to their stronger deleterious effects, whether transposition was kept constant or not. Models including preference for TEs to insert in regions of high recombination (panels C and D) produced very similar results for this summary statistic.

The density of polymorphic insertions was higher in regions of high recombination for short TEs across all simulations, but the difference was even more pronounced when preference for regions of high recombination was added to the model (panels C and D). The trend was reversed for long TEs (panel A), but including preference for high recombination again

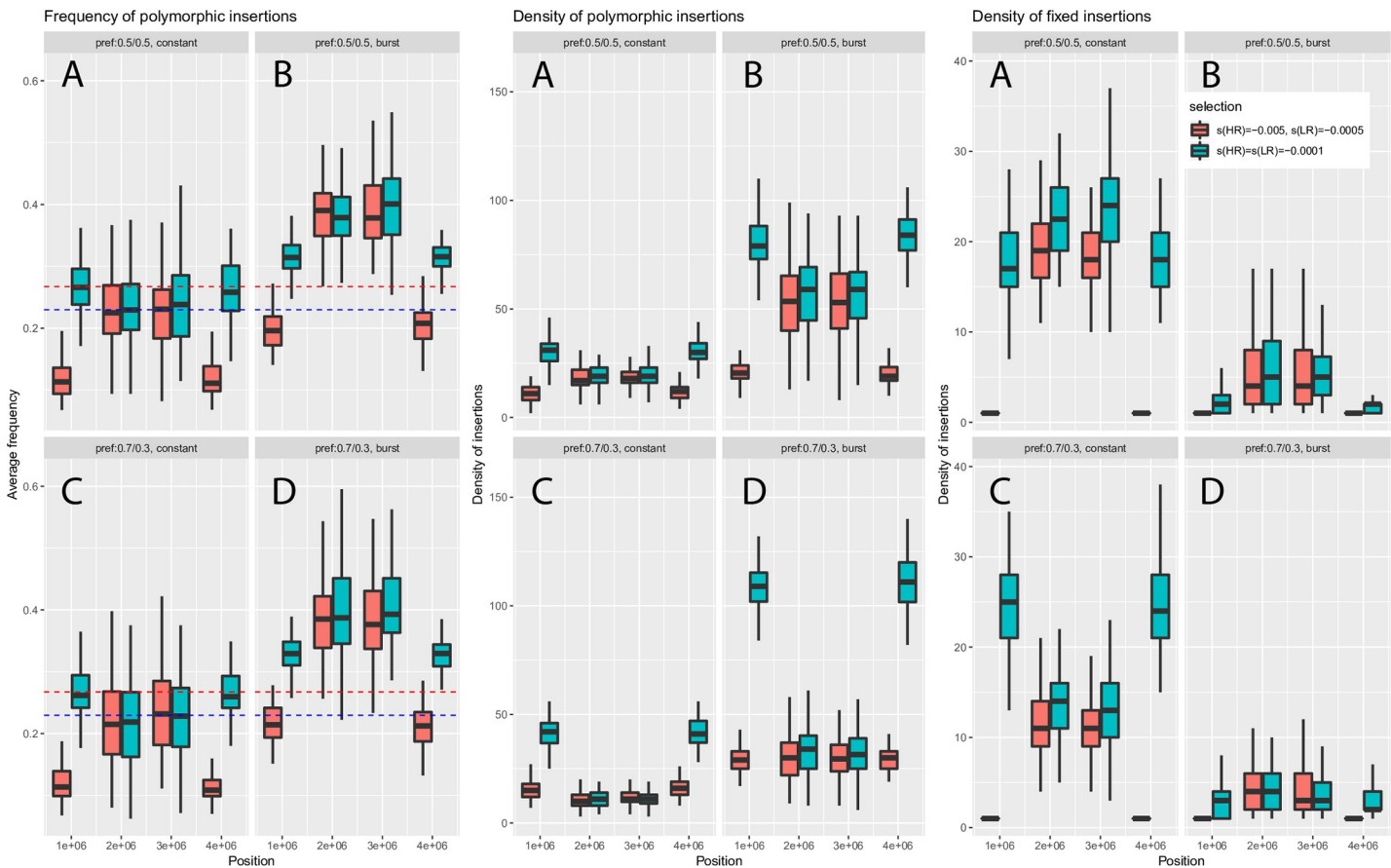

**Fig 9. Summary of simulations of TEs using SLiM3, using parameters realistic for the NEF cluster. Eight diploid individuals were sampled to mimic our sampling scheme.** Boxplots correspond to the results obtained over 100 simulations of a 4Mb fragment, divided into three regions of 1, 2 and 1 Mb. The first and last Mb correspond to regions of high recombination and high density of functional coding sites (respectively 10 times and 2 times higher than in the 2Mb central region). Coefficients of selection and other parameters are scaled using an effective population size of 1000 instead of 1,000,000 to reduce computation time (see Methods). $2N_e s$ = -10 for 10% of non-coding sites and $2N_e s$ = -100 for 70% of coding sites. Blue and red dotted lines correspond to average derived SNP frequencies in regions of low and high recombination respectively. A: model with constant transposition and no preferential insertion; B: model with a burst of transposition. C and D are the same than A and B respectively, but include preferential insertion in regions of higher recombination such that 70% of new elements go into these regions. TEs that fall in coding regions are strongly deleterious (selection coefficient 2Ns = -2000).

led to a positive correlation between recombination rate and the summary statistic (panel C), since more insertions could replace the ones erased by selection. Models where a burst of transposition occurred gave the same trends (panels B and D), although preference for high recombination did not fully reverse the correlation (panel D).

The density of fixed insertions was lower in regions of high recombination than in regions of low recombination in models with no preference (panels A and B). This result was observed for both short and long TEs, although the effect was enhanced for long TEs due to their stronger deleterious effect in regions of high recombination. In models where preferential insertion in regions of high recombination was added however, a positive correlation with recombination rate was observed under a constant transposition rate, and differences were less marked in the case of a transposition burst (panels C and D).

Our observations remained valid under scenarios with varying proportions of point mutations under purifying and positive selection, and selective coefficients (S8 to S12 Figs). Stronger purifying selection (S8 Fig, $2N_e s$ = -400 in coding sequences, $2N_e s$ = -40 in non-coding sequences) led to results similar to the ones shown in Fig 9, but the density of polymorphic

TEs tended to be even lower in regions of low recombination for "long" elements, reducing the contrast with regions of high recombination (panels A and B). In the case of weaker purifying selection (S9 and S10 Figs), we observed little difference in TE frequencies and densities across windows for "short" elements, consistent with their near-neutral behavior. At last, we note that adding modest amounts of positively selected sites to this latter model with weak purifying selection restored partially the correlations observed with strong purifying selection alone (S11 and S12 Figs).

We compared these trends with our actual observations (summarized in Table 2), which are consistent with either strong purifying selection against new insertions through ectopic recombination or predominant effects of linked selection. For short nLTR-RTs, and particularly *CR1*, we observed correlations consistent with linked selection, similar to the simulations for short elements highlighted in panels A of Fig 9. A possible effect of preferential insertion may explain the weak correlations observed between the density of fixed elements and recombination for *L1* and *L2* (panels C). For long nLTR-RTs, correlations for the three statistics were consistent with simulations obtained for long elements with no preferential insertion (Fig 9, panels A and B). The same was observed for *Gypsy* elements. For long *L2*, the lack of strong correlation between the density of polymorphic elements and recombination may reflect a situation closer from the simulations presented in panels C and D, with some effect of preferential insertion and past burst of transposition. The same reasoning may be applied to LTR-RT elements such as *BEL*. For *Dirs*, observations matched expectations for long elements in simulations shown in panel C, suggesting both selection against ectopic recombination and preferential insertion in regions of high recombination. For SINEs and *Tc1/Mariner*, the observed correlations clearly matched simulations for short elements including linked selection and preferential insertion (panel C). This scenario is also likely for *Hobo* and *Helitron*, although their weak frequencies obscures correlations between average allele frequencies, density of fixed sites, and recombination. The same issue makes any interpretation of patterns observed for *Penelope* difficult.

Given the high frequencies observed for *SINE2* and particularly *hAT*, it is possible that lower transposition rates in more recent times have led to a situation intermediate between our constant transposition and ancient burst scenarios for short elements (panels C and D respectively), weakening correlations between average frequencies and recombination.

## Are TEs targeted by strong and recent positive selection in northern populations?

Because TEs can cause major regulatory changes, they may be recruited during local adaptation, especially in species encompassing a broad range of environmental conditions. If TE insertions were recruited during the recent colonization of northern environments, they should display a strong change in frequencies between the Floridian source and northern populations, and fall in regions displaying signatures of positive selection that can be detected through the use of SNP data. We first scanned all polymorphic insertions to identify a set of candidate TEs displaying high frequencies in Northern clusters and low frequency in Florida. We used two statistics to identify TEs that were potentially under positive selection, $X^TX$ and *eBPis* [41]. $X^TX$ is a measure of global differentiation that should be higher for markers displaying variation in allele frequencies that are not consistent with demographic expectations drawn from SNPs. *eBPis* is a complementary statistic that specifically contrasts frequencies between Floridian or Northern clusters. We identified a set of 34 insertions that were in the top 1% for both *eBPis* and $X^TX$ statistics and showed a shift of at least 0.5 in their frequency compared to all samples in Southern Florida. We then filtered out insertions that did not fall

in a set of candidate windows displaying consistent signals of selection across three different approaches (diploS/HIC [42], BAYPASS [41] and LSD [43]; see Methods). Seventeen and 15 TE insertions overlapped with windows in the top 10% for the LSD score and BAYPASS score respectively (Table 3). Eight TE insertions fell in a window classified as a sweep by diploS/HIC. A total of six insertions were found in candidate windows for selection in all three tests (Table 3), four of them found within three distinct genes, a neurexin, *PTBP3*and *TCF-20-201*.

## Discussion

Using empirical data in a model species harboring a large diversity of active TEs as well as simulations, we investigated the relative impact of selective and non-selective factors on the population dynamics of all the main TE categories active in vertebrates. We tested how the combination of linked selection in the host, direct selection against TEs and changes in transposition rate may explain heterogeneous TE frequency and abundance along the genome. By comparing the diversity of several of the most common TE categories found in vertebrates within the same organism, we clearly demonstrate that the interaction between these processes lead to sometimes drastically different outputs, even under a shared demographic history. It may be possible to disentangle these different processes using information about elements length, genomic location and frequency.

### Demography shapes TE diversity across populations

We observed a clear effect of genetic drift on TE diversity across the genetic clusters examined in this study. Past work on green anole demography clearly showed that the GA and CA clusters expanded recently after a bottleneck when populations contracted to reach about 10% of their ancestral sizes [30]. This is associated with a reduction in the total number of polymorphic insertions found in these populations (Fig 1, Table 1), but also in an increase in the number of fixed elements compared to Floridian samples. Across families and clades, there were between 5 and 20% more fixed insertions in northern samples than in Florida (Table 1). This is a classical expectation: under a bottleneck, rare mutations frequently go extinct while frequent ones tend to reach fixation, leaving an excess of mutations at intermediate frequencies [44]. Fixation may also be facilitated by relatively less efficient selection due to lower effective population size, reducing $N_e s$. The strong impact of demography on TE abundance and frequencies has also been observed in a broad range of species and TE families, such as SINEs, nLTR-RTs, *Ac*-like elements and *Gypsy* in several species of *Arabidopsis* [11,45]. In *Drosophila subobscura*, recent bottlenecks may also explain the unusually high frequencies of *Gypsy* and *bilbo* elements [46].

### Linked selection affects TE frequency, but does not fully explain TE density

We obtained intriguing results for SINEs, DNA transposons such as *Tc1/Mariner*, and short nLTR-RTs. Under the ectopic recombination hypothesis [36,47], which is usually invoked to explain genome-wide patterns of TE diversity, TEs tend to be removed from regions of high recombination through purifying selection. Such correlations have been commonly observed for several TE families in fruit flies and other vertebrates [7,36,48,49]. This should lead to negative correlations between recombination and TE diversity or abundance, assuming constant transposition. Instead, we observe a positive correlation between recombination and average frequency and density of polymorphic elements. Such positive correlation between allelic diversity and recombination is however a well-known feature of so-called "linked selection" [14,17]. Haplotypes harboring deleterious mutations tend to be longer in regions of low recombination, and competition between them reduces the efficacy of selection [17]. Similarly,

**Table 3. Summary of the 34 TE insertions candidate for positive selection.** None but two of these insertions were found in *A. allisoni* and *A. porcatus*. For each insertion, the putative length estimated by MELT is provided. LSD scores, median *eBPis* for SNPs (obtained with BAYPASS), and diploS/HIC classifications for windows containing the focal TE are given. Windows that are above the 90% percentile or classified as sweeps are highlighted with an asterisk.

| Chromosome | Position | Clade | TE putative length | Nearest gene | Distance to nearest gene | Frequency in outgroups (/4) | Frequency in Florida (/30) | Frequency in North (/22) | diploS/HIC classification | median eBPis (SNPs) | LSD score | Number of tests above 90% percentile/ classified as sweeps |
|---|---|---|---|---|---|---|---|---|---|---|---|---|
| 2 | 128671912 | ERV | 9190 | *rapgef6* | 17445 | 0 | 0 | 16 | **Hard**\* | **3.86**\* | **0.61**\* | 3 |
| 5 | 27319544 | CR1 | 1751 | *TCF20-201* | 0 | 0 | 1 | 21 | **Hard**\* | **6.38**\* | **0.88**\* | 3 |
| 1 | 260526442 | L1 | 1680 | *Neurexin* | 0 | 0 | 1 | 21 | **Soft**\* | **5.06**\* | **0.87**\* | 3 |
| 1 | 260803841 | LTR-RT | 859 | *Neurexin* | 0 | 0 | 1 | 21 | **Soft**\* | **6.40**\* | **0.93**\* | 3 |
| 2 | 57599175 | CR1 | 3374 | *PTBP3* | 0 | 0 | 0 | 18 | **Hard**\* | **4.25**\* | **0.72**\* | 3 |
| 4 | 149706919 | CR1 | 1434 | *5S_rRNA* | 137822 | 1 | 2 | 20 | **Hard**\* | **5.06**\* | **0.83**\* | 3 |
| 4 | 153568233 | Hobo | 1957 | *COL20A1* | 0 | 2 | 3 | 22 | neutral | **3.10**\* | **0.74**\* | 2 |
| 6 | 80481846 | Gypsy | 6269 | *PRRG2* | 0 | 0 | 0 | 19 | neutral | **3.86**\* | **0.77**\* | 2 |
| 2 | 57632946 | Helitron | 539 | *PTBP3* | 20057 | 0 | 0 | 16 | neutral | **3.88**\* | **0.70**\* | 2 |
| 4 | 110324469 | Dirs | 5787 | *TESK2* | 7154 | 0 | 4 | 22 | neutral | **4.71**\* | **0.58**\* | 2 |
| 1 | 260754973 | CR1 | 1807 | *ENSACAG00000005710* | 0 | 0 | 2 | 22 | neutral | **4.78**\* | **0.90**\* | 2 |
| 1 | 89643089 | BEL | 718 | *TNS1* | 170148 | 0 | 0 | 13 | neutral | **4.88**\* | **0.66**\* | 2 |
| 5 | 26218266 | Gypsy | 1150 | *L3MBTL2* | 0 | 0 | 1 | 19 | neutral | **5.36**\* | **0.73**\* | 2 |
| 5 | 133673963 | BEL | 718 | *5S_rRNA* | 61322 | 0 | 0 | 14 | **Hard**\* | 1.87 | 0.55 | 1 |
| 6 | 64325965 | CR1 | 921 | *EFTUD2* | 0 | 0 | 1 | 18 | NA | 0.65 | **0.58**\* | 1 |
| GL343263 | 1240245 | Penelope | 2665 | *PPL* | 0 | 0 | 0 | 21 | NA | 0.91 | **0.82**\* | 1 |
| 4 | 10015690 | CR1 | 4292 | *ENSACAG00000030455* | 72218 | 0 | 6 | 16 | NA | **4.16**\* | 0.38 | 1 |
| 1 | 182379221 | LTR-RT | 859 | *SERINC1* | 20124 | 0 | 0 | 12 | neutral | 1.29 | **0.56**\* | 1 |
| 4 | 10794679 | Gypsy | 6279 | *NUDCD1* | 0 | 0 | 2 | 22 | neutral | 1.83 | **0.62**\* | 1 |
| 5 | 45604395 | L2 | 880 | *PKD2* | 0 | 0 | 4 | 20 | neutral | **3.89**\* | 0.49 | 1 |
| 3 | 104260103 | Gypsy | 969 | *NAXD* | 159686 | 0 | 0 | 12 | **Soft**\* | 1.41 | 0.32 | 1 |
| GL343312 | 1468585 | Gypsy | 6277 | *ENSACAG00000029426* | 3958 | 0 | 1 | 18 | NA | 1.22 | 0.35 | 0 |
| GL343285 | 1809184 | Gypsy | 301 | *ENSACAG00000027710* | 2326 | 0 | 0 | 12 | NA | NA | NA | 0 |
| 3 | 24577145 | Dirs | 32 | *DOCK10* | 0 | 0 | 0 | 12 | neutral | 0.29 | -0.09 | 0 |
| 5 | 133150214 | Penelope | 2665 | *ENSACAG00000029401* | 21423 | 0 | 0 | 12 | neutral | 0.71 | 0.29 | 0 |
| 1 | 197986056 | Gypsy | 888 | *ENSACAG00000026030* | 8212 | 0 | 0 | 15 | neutral | 0.89 | 0.38 | 0 |
| 3 | 63538365 | Gypsy | 1174 | *ADAM12* | 9569 | 0 | 1 | 18 | neutral | 1.29 | 0.05 | 0 |
| 4 | 67205243 | CR1 | 1944 | *GABBR2* | 0 | 0 | 3 | 21 | neutral | 1.42 | 0.46 | 0 |
| 5 | 67232158 | LTR-RT | 859 | *HYAL4* | 35781 | 0 | 1 | 21 | neutral | 1.49 | 0.08 | 0 |
| 5 | 106252550 | Penelope | 2665 | *SGCZ* | 11505 | 0 | 0 | 15 | neutral | 2.08 | 0.04 | 0 |
| 4 | 83252786 | Dirs | 5672 | *VAV3* | 0 | 0 | 0 | 13 | neutral | 2.10 | 0.29 | 0 |
| 3 | 24593897 | BEL | 718 | *DOCK10* | 0 | 0 | 0 | 12 | neutral | 2.11 | 0.30 | 0 |
| GL343198 | 969347 | Dirs | 108 | *RAB11FIP4* | 0 | 0 | 0 | 14 | neutral | 2.24 | 0.17 | 0 |
| 2 | 53454989 | Hobo | 1914 | *ENSACAG00000032857* | 35068 | 0 | 0 | 13 | neutral | 2.67 | 0.13 | 0 |

the local reduction in diversity that comes with selective sweeps extends over longer genomic distances in regions of low recombination. Altogether, this leads to an effect similar to a local reduction of effective population sizes in regions of low recombination, reducing diversity and increasing the odds that deleterious alleles reach fixation. We note that these elements tend to be quite short in length, which may make them nearly neutral, and therefore more likely to be impacted by linked selection.

While some work has been done in examining whether Hill-Robertson interference between elements may increase the number of fixed insertions in regions of low recombination [16], there is not any study (to our knowledge) that examined the allele frequency spectrum of TE insertions under linked selection. In addition, the latter study considered only TE insertions and did not incorporate background selection or sweeps on SNPs. Our simulations suggest that linked selection may lead to positive correlations between polymorphic TE frequency and abundance: polymorphic TEs would stochastically reach frequencies of 0 or 1 at a faster rate in regions of lower recombination. This would therefore lead to a rise in the number of polymorphic TEs and average TE frequencies as recombination increases, but also to a reduction in the number of fixed TEs (as expected in the case of Hill-Robertson interference).

Unlike the ectopic recombination and linked-selection hypotheses, preferential insertion in regions of high recombination and open chromatin does predict a positive correlation between recombination rates and TEs density. This mechanism has been proposed to explain why LINEs and LTR-RT may be more abundant in regions of high recombination in *Ficedula* flycatchers and the zebra finch [8]. In the case of LTR-RTs, it may also be that higher recombination rates increase the frequency of solo-LTR formation, limiting their deleterious effects. It is commonly observed for several retrotransposons in a variety of species [9,50,51]. However, in humans, *L1* may actually not display strong preference for open chromatin and is more constrained by local replication timing [52,53]. In the green anole, nLTR-RTs and LTR-RTs do not display strong evidence of preferential insertion in regions of high recombination, which tend to harbor less fixed elements. We note that these families may be older in birds than in the green anole, having accumulated between 55 and 33 Mya [54], while a substantial proportion of these elements display less than 1% divergence from their consensus in green anoles (see repeat landscape at http://www.repeatmasker.org/species/anoCar.html, last accessed 25/03/2020). It is therefore possible that purifying selection has had more time to remove the most deleterious insertions in birds, increasing the signal of preferential insertion that may be masked in the green anole. It may also be that LTR-RT elements produce more frequently solo-LTR in birds than in lizards, which would make them less deleterious and more subject to drift and linked selection. Further studies at finer genomic scales will be helpful to precisely quantify how local genomic features impact TE abundance.

Our simulations suggest preferential insertion would probably not produce higher average TE frequencies in regions of high recombination. We interpret this as the fact that preferential insertion is analog to locally higher mutation rates for nucleotides: while this may affect local SNP density along the genome, it should have little effect on the shape of the allele frequency spectrum under mutation-drift equilibrium (under the assumption of infinite sites which should hold for low mutation or transposition rates [55]).

We therefore propose that SINEs, *Tc1/Mariner* and most short elements are under the influence of linked selection and preferentially insert into regions of high recombination, possibly because these are more likely to be associated with an open chromatin state. Indeed, combining these mechanisms in our simulations produced correlations matching our observations for SINEs and *Tc1/Mariner*. The average frequency of these elements was quite close from average derived SNP frequency. It is therefore unlikely that strong purifying selection acts against these elements (Figs 4, 5, 6 and 9).

For short nLTR-RTs, the negative correlation between recombination rate and the density of fixed elements may reflect a residual effect of stronger purifying selection in regions of high recombination, and/or weaker preference for regions of open chromatin. In the case of short *L1*, we observe a positive correlation between the density of polymorphic elements and recombination rate, but this correlation is weak when examining the density of fixed elements or average frequency. We note however that *L1s* are substantially longer than other LINEs (Fig 8; average length of 3700 bp, 1194 bp and 1448 bp for *L1*, *L2* and *CR1* respectively for all insertions in the dataset, Wilcoxon tests, all $P<2.2.10^{-16}$), which limits our power to study short elements.

## Combination of bursts of transposition and linked selection leaves a specific signature

Sudden bursts of transposition are common in TEs, and have been documented in a variety of species [56–60]. This idiosyncrasy limits direct comparisons between TEs and SNPs, since mutation rates are usually considered constant for the latter. A general prediction is that the average frequency of elements should increase with their age, which is observed in *Drosophila* [48]. Our simulations also suggest that the positive correlation between average TE frequency and recombination rate observed for weakly deleterious TEs could be weakened and even reversed in the case of a sufficiently old transposition burst. This is due to the fact that the rarest elements have already been eliminated through drift, and the effects of linked selection lead to a faster accumulation of elements at high frequency in regions of low recombination.

We found that elements such as *hAT* and *L2* had substantially higher average frequencies, even higher than derived SNPs. For these two elements, correlations between their average frequencies and recombination rate were quite weak, even when considering only short *L2* that should be the least deleterious. This could reflect an intermediate situation compared to the extreme scenarios illustrated in Fig 9, such as multiple waves of transposition, or a younger burst than the one modeled, that may obscure correlations by flattening average allele frequencies. Examining the spectrum from more individuals may have the potential to reveal irregular transposition since local peaks in the spectrum should correspond to the age of each burst.

On the other hand, DNA transposons such as *Helitron* and *Hobo* are at very low frequencies, with almost no fixed insertion, but are more abundant in regions of high recombination. This pattern could be explained by a recent burst of transposition associated with weak purifying selection. Whether these elements share the preference of other DNA transposons for regions of high recombination remains difficult to assess due to the lack of fixed insertions.

## Strong purifying selection against Dirs, LTR-RTs and long nLTR-RTs

There is evidence that strong purifying selection acts on *Dirs*, LTR-RTs and long nLTR-RTs: their average frequency is generally lower than the one of derived SNPs. Recent bursts of transposition alone may also be responsible for an excess of young, therefore rare, alleles [61]. While this seems clearly the case for *Gypsy* elements, which display many singletons and seem to be less impacted by recent demography, we also found evidence for lower average TE frequency and density of fixed TEs in regions of high recombination for long nLTR-RTs and LTR-RTs. According to our simulations, such a correlation can only be obtained through stronger purifying selection in regions of high recombination, consistent with the ectopic recombination model. For all LTR-RTs (except *Gypsy*) and long *L2*, we observed weak and even positive correlations between recombination rate and the density of polymorphic elements. This may reflect some preference for regions of high recombination compensating the loss of polymorphic elements through selection.

These results suggest that LTRs and long nLTR-RTs may be more harmful in regions of high recombination, which are also richer in functional elements. Assuming that our simulations are reasonably close from the actual processes taking place in the green anole, $N_e s$ against these elements is likely high, and possibly higher for elements with very low frequencies such as *Dirs*, *Gypsy* or *BEL*. Full-length LTR-RTs are very long elements (~5,000bp) in the green anole, that may be strongly deleterious under the ectopic recombination model. In addition, they harbor regulatory motifs that may increase their deleterious effects near coding regions. The length of an element seems to be strongly correlated with its impact on fitness, since short LINEs display a weakening and even a reversal of correlations with recombination rate. These results are consistent with the ectopic recombination hypothesis, since longer elements are more likely to mediate ectopic recombination events [7].

## Strong recent positive selection on TEs is rare

Recent colonization of northern climates by the green anole may have been an opportunity for domestication of TEs, either through adaptation to the new selective pressures encountered or selection on dispersal promoting colonization of the new environment [62]. We did not find strong evidence that TEs be involved in adaptation in the northern populations. Only a few TEs displayed substantial differences in frequencies between northern and Floridian clusters. We found in total four elements that are serious candidate for positive selection, falling in introns of a neurexin gene, *PTBP3* and *TCF20-201*. *PTBP3* is involved in cell growth and erythropoiesis [63]. *Neurexins* are involved in the neurotransmitter release [64], while *TCF20-201* is a transcription factor associated with behavioral abnormalities [65,66]. While this suggests a potential impact on the nervous system and behavior, and echoes our findings from a previous study on positive selection in green anoles [62], further investigations are needed to formally validate the causal role of these elements and discard the possibility that they are only linked to a causal variant under selection. The fact that none of these elements was full-length (Table 3) makes substantial regulatory changes unlikely. Our results contrast with observations in *Drosophila*, where many TEs display steep clines in frequency that match environmental gradients and adaptive phenotypes [24,67,68]. Further investigations are needed to assess whether higher effective population sizes and more compact genome structure in *Drosophila* may explain higher rates of domestication.

There is a growing body of evidence that intrinsic properties of genomes (e.g. overdominance, Hill-Robertson effects, non-equilibrium demography) may lead to spurious signals of selection. We note that we are extremely stringent in our approach, requiring that at least three distinct tests of positive selection give a consistent signal, one of these tests explicitly incorporating demographic history in its implementation. While this could potentially limit our power to detect more subtle signals of positive selection (e.g. soft or partial sweeps), we caution against over interpreting results obtained from a single test, especially when demographic histories are complex. This is not to say that TEs are not more frequently involved over longer timescales: for example, TEs may be involved in speciation and morphological adaptation by shaping the *Hox* genes cluster in anoles [26]. Future studies on larger sample sizes may provide a more refined picture of the role of TEs in local adaptation.

## Perspectives on modelling TE dynamics

We created a simple model of TE evolution that incorporated variable purifying selection against TEs, bursts of transposition, preferential insertion of TEs in regions of high recombination, and linked selection. While this model was designed as a way to illustrate how different combinations of parameters may impact correlations for the three main statistics examined in

this work, this constitutes a template for future, more detailed studies of TE evolution. For example, SLiM3 allows the incorporation of detailed maps of genomic features, complex demographic histories, multiple modes of selection, or asexual reproduction. This should facilitate the interpretation of TE diversity in species for which a reference genome is available, and improve our understanding in model species for which extensive genomic information exists. Simulated data could be used in an ABC-like approach [13], or to train machine learning algorithms [69]. Such approaches may have the power to directly quantify for each TE clade the strength of purifying selection and how other processes such as linked selection and transposition process may interact.

## Materials and methods

### Sampling and SNPs calling

Liver tissue samples from 27 *Anolis carolinensis* individuals were collected between 2009 and 2011 (Tollis et al. 2012), and *A. porcatus* and *A. allisoni* were generously provided by Breda Zimkus at Harvard University. Whole genome sequencing libraries were generated from these samples following the laboratory and bioinformatics procedures already presented in [30] and detailed in S1 Text. Sequencing depth was comprised between 7.22X and 16.74X, with an average depth of 11.45X. SNP data included 74,920,333 variants with less than 40% missing data. Sequencing data from this study have been submitted to the Sequencing Read Archive (https://www.ncbi.nlm.nih.gov/sra) under the BioProject designation PRJNA376071. We excluded one individual with low depth of coverage from subsequent analyses due to its large amount of missing data.

### Calling TEs

We used the Mobile Element Locator Tool (MELT) to identify polymorphic insertions in the green anole genome [70]. This software performs well in identifying and genotyping polymorphic TEs in resequencing data of low and moderate coverage (5-20X), using TE consensus sequences (available at https://github.com/YannBourgeois/SLIM_simulations_TEs) to identify reads mapping to both the reference genome and the consensus. We followed the same pipeline used in previous studies [12,13], but included several clades of transposable elements covering SINEs, nLTR-RT LTR-RT and DNA transposons, using all available consensus sequences available on Repbase [71] to call TEs. Note that MELT can estimate the most likely breakpoints, insertion length, and strand for each insertion. We followed the MELT-SPLIT pathway, which consists of four main steps. First, TEs are called for each individual separately (IndivAnalysis). Then, calling is performed at the scale of the whole dataset to improve sensitivity and precision when estimating breakpoints and insertion length (GroupAnalysis). This information is then used to genotype each individual (Genotype), after what a VCF file is produced that lists all polymorphisms (makeVCF). To draw an accurate estimate of TE frequency spectra, we also used MELT-DELETION to identify polymorphic insertions found in the reference but not in all sequenced individuals. We called polymorphic TEs for each clade within the four main categories, using a threshold of 5% with the consensus sequence to attribute an element to a specific clade. The resulting VCF files were then merged for each of the four main categories considered. In case of a possible duplicate call (*i.e.* when two insertions were found at less than 2000bp from each other), only the insertion with the lowest divergence was kept. In case of equal divergence, the element with the highest calling rate was kept. We focused on TEs insertions with no missing data. While we acknowledge that these filters may be quite stringent, they should not have any impact on correlations with intrinsic genomic features and demography.

## Correlations with genomic features and SNPs statistics

From the MELT output, we extracted information about the frequency of each insertion in each of the five genetic clusters found in the green anole, using the option—counts in VCFTOOLS (v0.1.14) [72]. We also estimated the number of heterozygous sites for each individual using the—012 option in VCFTOOLS. For nLTR-RTs, we extracted the length of each insertion using shell scripts. We counted the number of insertions, and the proportion of private and shared alleles for each clade using R scripts [73].

We also investigated how TE diversity correlated with intrinsic features of the genome such as the recombination rate, and statistics related to demographic processes. We focused on three commonly used statistics to describe TE diversity in each genetic cluster: the density of polymorphic TEs along the genome, the density of fixed TEs along the genome, and the average frequency of polymorphic TEs in the host's population. Note that we do not include TEs that are fixed in all 29 samples since our interest in on the most recent population dynamics. We averaged TEs frequencies and densities over 1Mb windows, a length chosen to recover enough TEs even at the clade level, while limiting the effects of linkage disequilibrium and autocorrelation between adjacent windows. Windows with no TEs or found on scaffolds not assigned to any of the six main autosomes were excluded. To estimate average TE frequencies, only windows with at least three polymorphic insertions were used. We also extracted the average effective recombination rate $\rho = 4N_e r$ in the NEF clade estimated by LDHat (v2.2) [74] in a previous study, with $N_e$ the effective population size and $r$ the recombination rate between two adjacent sites (see S1 Text and [30] for details). This population was chosen since it has the largest sample size and has a large, stable effective population size. This rate was divided by another estimator of the effective population size, the average number of pairwise differences ($\theta_\pi = 4N_e\mu$, $\mu$ being the mutation rate per base pair), to obtain an estimate $r/\mu$ less sensitive to local reductions in effective population sizes due to linked selection. Relative and absolute measures of differentiation such as $d_{XY}$ and $F_{ST}$ were also computed over 1 Mb windows, as well as the average frequency of derived SNPs in green anoles, using the two outgroups *A. porcatus* and *A. allisoni* to determine the derived alleles. These last statistics were obtained using the package POPGENOME (v2.7.5) [75]. Correlograms summarizing correlations between these summary statistics, TE frequencies, and TE densities for the four main orders were obtained using the R package corrplot. Significance and strength of correlations were assessed using Spearman's rank correlation tests. For plots of correlation, regression lines and their confidence intervals were added to improve visibility with the function geom_smooth in ggplot2 (v3.2.1) [76], using a Gaussian model for TE frequencies and a Poisson model for TE densities (which are counts per window).

## SLiM3 simulations

In order to clarify how factors such as linked selection, bursts of transposition and preferential insertion of TEs may impact the three statistics examined in this study, we performed simulations using the forward-in-time simulator SLiM (v3.3.3) [40]. We modified a preexisting recipe (14.12) provided by Benjamin Haller and Philipp Messer. We simulated a 4Mb genomic fragment with parameters such as effective population size, exon density, mutation and recombination rates that were realistic for green anoles (S13 Fig). We simulated 8 diploid individuals drawn from a stable population with a $N_e$ of one million diploid individuals, similar to the NEF clade (S1 Fig). The mutation rate for nucleotides was set at $2.1.10^{-10}$ mutation/generation/site [28]. To simulate the effects of linked selection, we set the recombination rate at $2.10^{-10}$ /generation on the first and last Mb of the fragment, and at $2.10^{-11}$ /generation in the 2Mb between. These rates encompassed those estimated with LDHat in previous studies

[30,62]. Because regions of higher recombination tend to display more exons (S6 Fig), we assigned to regions of low and high recombination 10,000 bp and 20,000 bp of coding sequences per Mb respectively. We simulated 160 bp exons (close to the average length of exons in the green anole) that were randomly placed until the desired density was reached. To explore the effects of linked selection due to deleterious and positively selected sites, we varied the proportion of nucleotide mutations under selection. In exons, we kept the proportion of new deleterious point mutation at 70% in all simulations, which seems reasonable given that dN/dS in anoles are about 0.15 [77], suggesting that most substitutions at non-silent sites are deleterious (see box 2 in [78]). To obtain the results shown in Fig 9, we assumed that deleterious mutations in exons would display a strong effects on fitness, with $2N_e s$ = -100 ($s = 5.10^{-5}$). Of all new point mutations in non-coding regions, 10% were deleterious with $2N_e s$ = -10 ($s = 5.10^{-6}$). There is not much information about the fitness effects of new mutations in non-coding sequences in vertebrates in general, and the green anole in particular. However, our estimate seems conservative given that in mice and humans, about 20–40% of mutations in conserved regions may have an $s > 3.10^{-4}$ [79]. To explore further how varying selective coefficients may impact our results, we examined results from simulations with $2N_e s$ = -40 or $2N_e s$ = -1 in 10% of non-coding sites, and $2N_e s$ = -400 or $2N_e s$ = -10 in coding regions (S8 and S9 Figs). We also examined a case with almost no purifying selection on nucleotides, with only 1% of non-coding sites being under purifying selection (S10 Fig). We acknowledge that positive selection may also play a major role in reducing diversity, and also explored how adding positive selection to the latter model with little purifying selection on nucleotides may restore the correlations we observed with strong purifying selection. We added positively selected substitutions, with 1% and 5% of new substitutions in coding regions with $2N_e s$ = +10 (0.1% and 0.5% in non-coding regions, S11 and S12 Figs respectively)We assumed that there are 10 TE progenitors for a given TE clade in the whole genome that can jump and insert at $P = 1.10^{-3}$ elements/generation/genome at a constant rate, a value chosen to reflect known transposition rates in vertebrates and which produced a number of TEs close from our empirical observations for individual TE clades. This gave a probability of insertion in the 4Mb region of $P$ x 4 / 1780, since the green anole genome is 1.78 Gb long. We also modelled bursts of transposition where $P$ was set 100 times higher, but with transposition occurring only during a lapse of 100,000 years, starting 1,000,000 years ago. Note that in that latter case, TE insertions do not reach transposition-selection-drift equilibrium. Half of the newly generated elements were considered "short" and under weak purifying selection, with $2N_e s$ = -0.1. The other half were considered "long", and had a stronger impact on fitness when falling in regions of high recombination ($2N_e s$ = -10) than in regions of low recombination ($2N_e s$ = -1). The justification for this is that long elements have a higher probability of mediating deleterious ectopic recombination events and those events are more likely to occur in regions of high recombination. Both long and short TEs falling in coding sequences were considered strongly deleterious ($2N_e s$ = -2000).To improve the speed of simulations, we modelled a population of size $N_e$ = 1000 diploid individuals, and rescaled all parameters accordingly: mutation, recombination and rates of insertion were multiplied by a factor 1000, and times in generation and selection coefficients divided by the same factor. Simulations were run over 20,000 generations to ensure that mutation-selection-drift balance was achieved for nucleotide mutations.

To account for the potential preference of elements to insert in regions of high recombination, which tend to be gene rich and are often associated with open chromatin [8,80], we also added a preference bias $Q$ which could take the values 0.5 (TEs were as likely to insert in regions of low recombination than in regions of high recombination) or 0.7 (in that case, 70% of TEs jumping into the 4Mb region inserted in regions of high recombination and 30% in regions of low recombination). Note that values for selection coefficients and preferential

insertion were chosen to better visualize the trends that we observed across a range of other combinations, and because they produced results close from our empirical observations. The scripts used to simulate these data are available on Github (https://github.com/YannBourgeois/SLIM_simulations_TEs), and can be reused to explore in more details other combinations of parameters.

## Overlap with scans for positive selection

We used the approach implemented in BAYPASS (v2.1) [41] to detect TEs displaying high differentiation in northern populations. Overall divergence at each locus was first characterized using the $X^TX$ statistics, which is a measure of adaptive differentiation corrected for population structure and demography. Briefly, BAYPASS estimates a variance-covariance matrix reflecting correlations between allele frequencies across populations, a description that can incorporate admixture events and gene flow. This matrix is then used to correct differentiation statistics. BAYPASS offers the option to estimate an empirical Bayesian *p*-value (*eBPis*) and a correlation coefficient, which can be seen as the support for a non-random association between alleles and specific populations. BAYPASS was run using default parameters under the core model and using the matrix inferred from SNP data in [62]. We considered a TE as a candidate for selection in northern populations when belonging to the top 1% $X^TX$ and 1% *eBPis*, and if the difference in frequency with Florida was at least 0.5.

We compared our set of candidate TEs with the results obtained from a previous study on positive selection in the same northern populations [62]. Briefly, three different methods were applied and their results compared. We first used diploS/HIC [42], which is a machine-learning approach that uses coalescent simulations with and without selection to estimate which genomic regions are more likely to be under selection. This method has the advantage of incorporating past fluctuations in population sizes, which may reduce the number of false positives due to demography. We also used LSD [43], an approach that compares genealogies along genomic windows and detects those harboring short branches in the focal population compared to its sister clades, a signal of disruptive selection. At last, we also used BAYPASS on SNP data. Further details can be found in S1 Text and [62]. The set of candidate TEs for selection was compared with the set of candidate windows for positive selection and the intersection was extracted using BEDTOOLS (v2.25.0) [81].

## Supporting information

**S1 Fig. Summary of population structure and environmental variation in green anoles (see [30] for further details).** A: RAxML phylogeny on one million random SNPs. B: Demographic evolution of the five genetic clusters of green anoles reconstructed by SMC++ [82]. C: Sampling locations used in this study. Units for temperature are in tenth of Celsius degrees. D: PCA over environmental variables (BIOCLIM data) for the locations used in this study. Larger dots highlight the northern clades (GA and CA) and their sister Floridian clade (NEF). (PDF)

**S2 Fig. Allele frequency spectra for nLTR-RTs belonging to all five genetic clusters identified in the green anole.** (PDF)

**S3 Fig. Allele frequency spectra for SINEs belonging to all five genetic clusters identified in the green anole.** (PDF)

**S4 Fig. Allele frequency spectra for LTR-RTs belonging to all five genetic clusters identified in the green anole.**
(PDF)

**S5 Fig. Allele frequency spectra for DNA-transposon s belonging to all five genetic clusters identified in the green anole.**
(PDF)

**S6 Fig. Plot of the correlation between exon density and scaled recombination rate.**
(PDF)

**S7 Fig. Truncation of LINE elements that are assigned unambiguously to their consensus.**
Left: position of the start of an element relative to its consensus, reflecting 5' truncation. Right: position of the end of an element relative to its consensus.
(PDF)

**S8 Fig. Summary of simulations of TEs using SLiM3.** Legend is the same as Fig 9. Parameters: $2N_es$ = -40 for 10% of non-coding sites and $2N_es$ = -400 for 70% of coding sites.
(PDF)

**S9 Fig. Summary of simulations of TEs using SLiM3.** Legend is the same as Fig 9. Parameters: $2N_es$ = -1 for 10% of non-coding sites and $2N_es$ = -10 for 70% of coding sites.
(PDF)

**S10 Fig. Summary of simulations of TEs using SLiM3.** Legend is the same as Fig 9. Parameters: $2N_es$ = -1 for 1% of non-coding sites and $2N_es$ = -10 for 70% of coding sites.
(PDF)

**S11 Fig. Summary of simulations of TEs using SLiM3.** Legend is the same as Fig 9. Parameters: Same as S10, but includes positive selection with $2N_es$ = 10 for 0.1% of non-coding sites and 1% of coding sites.
(PDF)

**S12 Fig. Summary of simulations of TEs using SLiM3.** Legend is the same as Fig 9. Parameters: Same as S10, but includes positive selection with $2N_es$ = 10 for 0.5% of non-coding sites and 5% of coding sites.
(PDF)

**S13 Fig. Graphical summary of SLiM3 simulation parameters.** We simulate a 4Mb fragment, assuming the following unscaled parameters (see Methods for details about scaling): a stable effective population size of 1 million individuals, a mutation rate of $2.1.10^{-10}$/year, high recombination in the first and last Mb ($r = 2.10^{-10}$ /year), low recombination in the 2 Mb in the middle ($r = 2.10^{-11}$ /generation). Linked selection is modelled by introducing 10% of deleterious mutations with $2N_es$ = -10 in non-coding regions and 70% of deleterious mutations with $2N_es$ = -100 in coding regions. We assume that there are 10 TE progenitors in the whole genome that can jump $P$ generations/genome (constant rate). We also model bursts of transposition where the probability of jumping is 100X higher, but transposition occurs during a lapse of 100,000 years, deviating from transposition-drift balance. We also add an insertion bias $Q$ to model preferential insertion in regions of high recombination.
(PDF)

**S1 Text. Supplementary Methods detailing the procedures used in previous studies to call SNPs, infer recombination rates and detect candidate regions for positive selection.**
(DOCX)

## Acknowledgments

We are grateful to Breda Zimkus from the Museum of Comparative Zoology Cryogenic Collection in Harvard and J. Rosado from the Herpetology Collection for providing the samples of *Anolis porcatus* and *Anolis allisoni*. We thank Marc Arnoux from the Genome Core Facility at NYUAD for assistance with genome sequencing. This research was carried out on the High-Performance Computing resources at New York University Abu Dhabi.

## Author Contributions

**Conceptualization:** Yann Bourgeois, Imtiyaz Hariyani, Stéphane Boissinot.

**Data curation:** Yann Bourgeois, Robert P. Ruggiero, Stéphane Boissinot.

**Formal analysis:** Yann Bourgeois, Imtiyaz Hariyani.

**Funding acquisition:** Stéphane Boissinot.

**Investigation:** Yann Bourgeois, Stéphane Boissinot.

**Methodology:** Yann Bourgeois, Robert P. Ruggiero.

**Project administration:** Yann Bourgeois, Stéphane Boissinot.

**Resources:** Robert P. Ruggiero, Stéphane Boissinot.

**Software:** Yann Bourgeois, Robert P. Ruggiero, Imtiyaz Hariyani.

**Supervision:** Stéphane Boissinot.

**Validation:** Yann Bourgeois.

**Visualization:** Yann Bourgeois.

**Writing – original draft:** Yann Bourgeois, Stéphane Boissinot.

**Writing – review & editing:** Yann Bourgeois, Robert P. Ruggiero, Imtiyaz Hariyani, Stéphane Boissinot.

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
