## [Decision Letter · Decision Letter 0]

2 Jul 2020

Dear Dr Bourgeois,

Thank you very much for submitting your Research Article entitled 'Disentangling the determinants of transposable elements dynamics in vertebrate genomes using empirical evidences and simulations' to PLOS Genetics. Your manuscript was fully evaluated at the editorial level and by three independent peer reviewers. The reviewers appreciated the attention to an important problem and all praise the study for its originality and important findings. However the reviewers raised several concerns about the current manuscript. Based on the reviews, we will not be able to accept this version of the manuscript, but we would be willing to review again a revised version. 

If you decide to revise the manuscript for further consideration at PLOS Genetics, please aim to resubmit within the next 60 days, unless it will take extra time to address the concerns of the reviewers, in which case we would appreciate an expected resubmission date by email to plosgenetics@plos.org.

[LINK]

We are sorry that we cannot be more positive about your manuscript at this stage. Please do not hesitate to contact us if you have any concerns or questions.

Yours sincerely,

Cédric Feschotte

Associate Editor

PLOS Genetics

Bret Payseur

Section Editor: Evolution

PLOS Genetics

Reviewer's Responses to Questions

**Comments to the Authors:**

Reviewer #1: I think this is a very interesting work shedding light on the dynamics of TEs in non-mammals vertebrates by combining real data with simulations. The authors provide the readers with the scripts to perform the simulations so I think this can also be very useful for the community. I have a few suggestions and questions.

Line 70. References should be added for the so called "alternative model"

Line 179. The authors used LINEs as a synomym of nLTRs-RT in the text while in the figures they use mostly (not always) nLTRs-RT. For consistency, I would always used the same nomenclature or specify that they are used interchangeably.

Figure 1. Consider plotting "private", "shared with green anole only" and "shared with outgroup" in that order.

Figure 2 will benefit from adding the class of TE nearby the letters, e.g. A)nLTR-RTs; B) SINEs etc...

Line 214. The impact of recombination on TE frequencies has been studied before, see for example Petrov et al 2011, Kofler et al 2012, Kapun et al 2020.

Line 226. It is not clear to the reader whether the "differentiation metrics" in figure 3 are calculated with SNPs or with TEs. The figure legends only mentions TEs but this is not clear in the text. Line 232 starts with "Linked selection should have a similar effect on TEs" which leds the reader to think that the previous metrics where calculated for SNPs. If that were the case, figure legend and the figure itself should be edited (Density of polymorphic insertions should be on top of the TE part of the figure).

Line 226. Moreover, is not clear whether results for the "differentiation metrics" refer to polymorphic, fixed, or both types of variants.

Line 253, text in this paragraph and Figure 4 nomenclature is misleading. For example, in Figure 1 DIRS are included in LTR-RT class while in Figure 4 the legend reads LTR-RT and DIRS as if they were two different categories.

Line 278. Mention in the text that the average frequency is for polymorphic insertions.

Line 279. Mention here that correlations for density are done separately for polymorphic and fixed insertions.

Line 298. The relationship between TE size and recombination applies to all TE classes not only to LINEs (see e.g. Petrov et al 2011).

Line 325. I am curious to know why the authors chose the percentage (10%) of the SNPs to be under negative selection. Also why they simulated a 1+2+1 region with high+low+high recombination instead of e.g. 2+2. Maybe these small details can be added to the material and methods?

Figure 9. Legend does not explain what A and B are in the figure. The discontinuos read and blue lines that appear in the first set of plots are not explained either.

Line 391. Reference to Gautier 2015 where XtX and eBPis is described.

Line 398 I would suggest to mention here the three approaches used to identify positive selection and reference their publications.

Table 2. I would suggest to not use the abbreviations for the dominant process (process is missing an "s" in the table) because there is enough space to add the process itself, which facilitates the reading. Some abbreviations might be used e.g. "purifying selection against ectopic recombination" could be "purifying sel ectopic rcb" or something similar if needed (It looks like there is enough space to add the whole information).

Line 430 Can the positive correlation between recombination rate and average frequency and density of polymorphic elements be explained by an enrichment of short lenght elements in the SINEs, and DNA transposons such as Tc1/Mariner groups of TEs? The correlation between recombination rate and average frequency is expected for long TE copies (see for example Petrov et al 2011).

Line 555 Although I do not disagree with the authors in their approach, I do think it might be way too stringent. How many TEs appear to be under positive selection with each individual test or according to two of the tests? I am curious to know these numbers, maybe they can be given and discussed.

Reviewer #2: In this paper, Bourgeois et al. investigated the dynamics of segregating and fixed transposable element copies in the genome of Anolis carolinensis. Different types of elements show contrasting patterns of correlations with recombination rate. This relationship may also vary between the fixed and the polymorphic elements within a given group. These results may be explained by demography, linked selection and/or preferential insertions. The results also show a lack of selection against short insertions and the authors used very careful approaches to identify limited signatures of adaptive TE insertions in recent evolutionary history.

The manuscript is well-written and certainly of interest to the readership of PLoS Genetics. However, some points require minor revision for clarity – especially related to the high density of functional sites in high-recombination regions and the regulatory potential of short vs. long TE insertions. My detailed comments are as follows (ordered by line number):

1. General comment on the usage of linked selection: Throughout the manuscript it is unclear if linked selection includes or excludes the effect of selection on the TEs themselves.

2. General comment on simulation: In SLiM 3 a recombination map is set up at the beginning of a run and each generation a single recombination map applies for all individuals (see page 566 of the SLiM manual). This could potentially be an issue here and something the authors should carefully consider.

3. Line 31: Please replace “TEs frequencies” with “frequency of TEs”.

4. Line 49: Please replace “he” with “the”.

5. Line 50: Please change “diversity” to “diversities”.

6. Line 68: How can expansion be independent of transposition? Through segmental duplications? A clarification of a stochastic expansion here would be good and separating genetic drift affecting TE copies and host.

7. Line 79: Is it correct to say “Hill-Robertson interference and hitchhiking”, considering that hitchhiking is a form of HRI?

8. Lines 101-102: Could you discuss here the presence of promoters and enhancers depending on TE structure and mechanism?

9. Line 150 and other appropriate sections: In the beginning of the Results and Figure 1, it would help the reader if the bottlenecked populations are explicitly mentioned or highlighted. Maybe a bracket around GA and CA as "Northern clades" would help? Please also mention which population the anole reference genome belongs to, in case the reader wonders about potential reference biases.

10. Line 154 and other appropriate sections: Conceptual question regarding cut-and-paste DNA transposons - how are allele frequencies dealt with for these?

11. Line 167: Please provide a reference for "low homoplasy of TE insertions".

12. Line 177: Please replace “population” with “populations”.

13. Line 179: Please insert "heterozygous" before "TEs".

14. Line 186-190: I admit it is not clear to me whether the high numbers of heterozygous TEs in outgroups are due to technical reasons (more non-reference insertions with more phylogenetic distance) or biological (e.g., reflecting differences in TE expansion or host population sizes)? I suggest double-checking for a potential role of coverage difference in the number of identified heterozygous TEs (i.e., more false negatives with lower coverage). If plotting the coverage per individual vs. the number of identified heterozygous TEs, is there a positive correlation?

15. Line 205: Can you comment on the size of LTRs (larger than LINEs) and the expectation that each LTR insertion (full-length and solo) should contain regulatory elements, while 5'-truncated LINEs should not contain these?

16. Line 235: What is meant with the phrase "purifying selection against elements through ectopic recombination"? Do you mean that TEs are selected against in regions where they would otherwise lead to ectopic recombination?

17. Line 235: Has it been proven that the rate of ectopic recombination is positively correlated with the population-effective recombination rate or crossover rate? Barrón et al. (2014; Population Genomics of Transposable Elements in Drosophila. Annual Review of Genetics, 48(1), 561–581. https://doi.org/10.1146/annurev-genet-120213-092359) states that it’s assumed. If you know of any source that have shown such a relationship, it would be helpful to the field to provide. Otherwise it would be good to clarify that it’s an assumption.

18. Line 244: Please insert "the variation in" before "their".

19. Lines 245-249 and other appropriate sections: Unclear what is meant in this paragraph - if there is a positive correlation of exon density and recombination rate, doesn't this higher density of functional elements explain the low density of LTRs and LINEs in these regions? Please revise this paragraph for clarity and consider adding this point in other appropriate sections.

20. Lines 250-252: Unclear what is meant with this paragraph - is the point here the similarity in densities between TE groups, or between host populations? Please revise this paragraph for clarity. One of the confusing aspects is the use of terms like "genetic groups" and "clades" in slightly different contexts across paragraphs.

21. Lines 272-275: What about the possibility that occasional cut-and-paste would reduce the frequency of (nearly) fixed DNA transposon insertions?

22. Line 276: See above comment #17.

23. Line 285: I agree that ectopic recombination would be a strongly deleterious effect of TEs in high-recombination regions, however, considering that exon density also correlates with recombination rate, what about the possibility that large TEs containing regulatory elements are particularly deleterious in regions with many genes?

24. Line 299: See comment #15, what if longer LINEs have regulatory elements in their 5' ends that shorter, truncated LINEs don't have?

25. Line 301: The assumption for this conclusion is that average TE copy age is uncorrelated to average TE copy length, however, this is not tested here and might not be true. The rate of deletion may also be correlated with recombination, see Nam and Ellegren (2012; Recombination Drives Vertebrate Genome Contraction. PLoS Genetics, 8(5), e1002680. https://doi.org/10.1371/journal.pgen.1002680). As a control, did you test whether any elements are 3'-truncated (instead of being 5’-truncated as expected from the insertion mechanism)?

26. Line 313: What does ">30% max length" mean?

27. Line 316: I advise the authors to interpret this more cautiously as recombination may create shorter elements through random deletion (Nam and Ellegren, 2012). This could be especially true for L2s which contain a recombination hotspot motif, at least in humans (McVean, 2010; What drives recombination hotspots to repeat DNA in humans? Philosophical Transactions of the Royal Society B: Biological Sciences, 365(1544), 1213–1218. https://doi.org/10.1098/rstb.2009.0299).

28. Lines 321-322 and other appropriate sections: Would it be possible to take into account the density of functional elements (exon density, for example)? After all, the strength of purifying and linked selection don't only depend on the recombination rate, but also on the distance of a TE insertion to a functional site under purifying selection.

29. Line 324: Here you write that you “built a simple model” though the core structure of your model greatly resembles (i.e. equal variable names with identical capitalizations) recipe 14.12 in SLiM3. I agree that you have modified the original recipe and created your own specific model which could motivate this kind of wording, but please also acknowledge the recipe creator.

30. Line 386: In line with an earlier comment, please consider whether the TE insertion of interest can actually contain a regulatory element or not (e.g., severely 5'-truncated LINEs).

31. Lines 452-454: I fear I don't follow here - wouldn't high recombination reduce the effect of Hill-Robertson interference? Could the reduced number of fixed TEs result from the high density of functional sites (exon density)?

32. Line 459: Note that the study cited was not able to differentiate between full-length LTRs and solo-LTRs, however, it is possible that frequent solo-LTR formation under high recombination rates will allow these to accumulate in high-recombination regions. If space allows, please consider mentioning this point here.

33. Lines 463-464: The point of LTR accumulation being ancient in birds is incorrect, as recently shown by two population-level studies (ref. 44 and Weissensteiner et al. 2020 The population genomics of structural variation in a songbird genus. biorxiv doi:10.1101/830356). Instead, most of the recent TE activity in songbirds appears to be LTR families. Please revise accordingly. What if all these LTRs are solo-LTRs in birds but less often so in anole?

34. Line 489: A reference to this statement would be helpful for the reader or some numbers from the present data.

35. Line 570: Please replace "details" with "detailed".

36. Line 592 and other appropriate sections: Two key points of the "Calling TEs" methods section should be mentioned in the Results/Discussion sections (because most studies on this topic should do this but don't): 1) that both reference and non-reference insertions were analyzed and 2) only insertions with no missing data for genotypes were analyzed.

37. Line 628: Unclear what densities and frequencies refer to here - relative to genomic distribution or to the sampled populations?

38. Line 655: Please briefly describe in the text here what parameters are realistic for anoles.

39. Line 664. I get 1.10-3.5 to be ~0.72. Is this really the selection coefficient intended?

40. Line 668: Is the assumption of 10 TE progenitors realistic given the many active families/subfamilies of TEs in anole?

41. Line 683: With the transposition-burst model, is mutation-selection-drift balance really achieved? Since mutations are only entering for a limited number of generations, this might seems unlikely.

42. Figure 2: Please consider showing AFS side-by-side barplots for easier readability.

Reviewer #3: This is an interesting study addressing the landscape of polymorphism and fixation of TEs, with accompanying simulations, to make inferences about the factors influencing TE polymorphism in natural populations of Anolis lizards.

I found the study interesting, and the conclusions generally well supported. However I do have some concerns and suggestions that I think are well worth considering in a revised version:

1) the section on the 'visual inspection of allele frequency spectra' could do with some statistical tests (e.g. Mann-Whitney tests). I realize there's probably sufficient data for these to be generally significant, but worth demonstrating

2) I think the introduction and various parts of the manuscript could do with a broader discussion of the effects of recombination on TEs. The intro focuses primarily on linked selection effects, but there are important alternatives that come up elsewhere in the manuscript, such as ectopic recombination and insertion preferences. Because recombination effects come up across the paper, I think a broader introduction to these issues is warranted. It would also be helpful to introduce alternative scenarios in the introduction, dependent on whether TE insertions are nearly neutral or subject to strong purifying selection.

3) Page 9, line 232- related to point 2 above, some of the predictions being made should be contextualized more. e.g. saying that 'linked selection should have similar effect on TEs as on SNPs' is accurate if we are talking about effectively neutral TEs and effectively neutral SNPs. Since this ends up being a part of the interpretation, I think it's important to make this clear in the setup to the test. You bring up the 'on the other hand' immediately after, but I think for clarity it's worth emphasizing here that this is if TEs are effectively neutral.

4) page 8, line 245- Unless I'm very much misunderstanding, I think the first sentence should read: 'The higher abundance of some TE categories in regions of HIGH recombination', not low? Otherwise I'm very confused about this, since regions of low recombination DO seem to show weaker deleterious effects due to lower exon density

5) page 10, line 276- related to points above, all of a sudden the ectopic exchange model is introduced, now ignoring the potential countervailing effects of linked selection, which now come up as an alternative explanation for SINEs and DNA elements, even though linked selection effects were brought up already earlier as the 'primary expectation for TE polymorphic densities. Again, I think it's helpful to spell out more clearly distinct expectations for when TEs are under direct purifying selection (either due to ectopic recombination or a higher density of functional elements) and when they are effectively neutral, and how this could drive alternative results. Presenting these as two (three?) alternatives throughout the results and intro will be extremely helpful.

6) I really liked the simulation results and their connection to the data. One concern I had though, is that Ns=10 is still pretty weak purifying selection, especially for thoughts about the average effect of ectopic recombination. Can the authors add one more case, of Ns=100?

7) I don't feel strongly, but from my sense 'lineage sorting' is usually not discussed in the context of linked selection, when we are not discussing two taxa and the effects of 'incomplete' or 'complete' lineage sorting. How about just referring to the effects of linked selection as 'shortening coalescent times'?

**Have all data underlying the figures and results presented in the manuscript been provided?**

Reviewer #1: Yes

Reviewer #2: **No: **Some data files underlying the figures would need to be added.

Reviewer #3: **No: **I would suggest that the TE insertion calls, and the data on TE frequencies and densities should be released in some way. Certainly the TE frequency/density info could be posted to Dryad, and the locations of insertions could be as well. VCFs are harder to release.

PLOS authors have the option to publish the peer review history of their article (what does this mean?). If published, this will include your full peer review and any attached files.

Reviewer #1: No

Reviewer #2: No

Reviewer #3: No

---

## [Editor Report · Decision Letter 1]

25 Aug 2020

Dear Dr Bourgeois,

We are pleased to inform you that your manuscript entitled "Disentangling the determinants of transposable elements dynamics in vertebrate genomes using empirical evidences and simulations" has been editorially accepted for publication in PLOS Genetics. Congratulations!

Yours sincerely,

Cédric Feschotte

Associate Editor

PLOS Genetics

Bret Payseur

Section Editor: Evolution

PLOS Genetics

Comments from the reviewers (if applicable):

**Data Deposition**

http://datadryad.org/submit?journalID=pgenetics&manu=PGENETICS-D-20-00628R1

**Press Queries**

---

## [Editor Report · Acceptance letter]

28 Sep 2020

PGENETICS-D-20-00628R1 

Disentangling the determinants of transposable elements dynamics in vertebrate genomes using empirical evidences and simulations 

Dear Dr Bourgeois, 

We are pleased to inform you that your manuscript entitled "Disentangling the determinants of transposable elements dynamics in vertebrate genomes using empirical evidences and simulations" has been formally accepted for publication in PLOS Genetics! Your manuscript is now with our production department and you will be notified of the publication date in due course.

With kind regards,

Matt Lyles

PLOS Genetics

On behalf of:
